# Proximal Curriculum for Reinforcement Learning Agents

**Georgios Tzannetos**                                            *gtzannet@mpi-sws.org*
*Max Planck Institute for Software Systems*

**Bárbara Gomes Ribeiro**                                         *bgomesr@mpi-sws.org*
*Max Planck Institute for Software Systems*

**Parameswaran Kamalaruban**                              *kparameswaran@turing.ac.uk*
*The Alan Turing Institute*

**Adish Singla**                                                 *adishs@mpi-sws.org*
*Max Planck Institute for Software Systems*

**Reviewed on OpenReview:** *https://openreview.net/forum?id=8WUyeeMxMH*

## Abstract

We consider the problem of curriculum design for reinforcement learning (RL) agents in contextual multi-task settings. Existing techniques on automatic curriculum design typically require domain-specific hyperparameter tuning or have limited theoretical underpinnings. To tackle these limitations, we design our curriculum strategy, PROCURL, inspired by the pedagogical concept of *Zone of Proximal Development* (ZPD). PROCURL captures the intuition that learning progress is maximized when picking tasks that are neither too hard nor too easy for the learner. We mathematically derive PROCURL by analyzing two simple learning settings. We also present a practical variant of PROCURL that can be directly integrated with deep RL frameworks with minimal hyperparameter tuning. Experimental results on a variety of domains demonstrate the effectiveness of our curriculum strategy over state-of-the-art baselines in accelerating the training process of deep RL agents.

## 1 Introduction

Recent advances in deep reinforcement learning (RL) have demonstrated impressive performance in games, continuous control, and robotics (Mnih et al., 2015; Lillicrap et al., 2015; Silver et al., 2017; Levine et al., 2016). Despite these remarkable successes, a broader application of RL in real-world domains is often very limited. For example, training RL agents in contextual multi-task settings and goal-based tasks with sparse rewards still remains challenging (Hallak et al., 2015; Kirk et al., 2021; Andrychowicz et al., 2017; Florensa et al., 2017; Riedmiller et al., 2018).

Inspired by the importance of curricula in pedagogical domains, there is a growing interest in leveraging curriculum strategies when training machine learning models in challenging domains. In the supervised learning setting, such as image classification, the impact of the order of presented training examples has been studied both theoretically and empirically (Weinshall et al., 2018; Weinshall & Amir, 2018; Zhou & Bilmes, 2018; Zhou et al., 2021; Elman, 1993; Bengio et al., 2009; Zaremba & Sutskever, 2014). Recent works have also studied curriculum strategies for learners in sequential-decision-making settings, such as imitation learning (where the agent learns from demonstrations) and RL (where the agent learns from rewards). In the imitation learning setting, recent works have proposed greedy curriculum strategies for picking the next training demonstration according to the agent's learning progress (Kamalaruban et al., 2019; Yengera et al., 2021). In the RL setting, several curriculum strategies have been proposed to improve sample efficiency, e.g., by choosing an appropriate next starting state or goal state for the task to train on (Wöhlke et al., 2020; Florensa et al., 2017; 2018; Racanière et al., 2020; Riedmiller et al., 2018; Klink et al., 2020a;b; Eimer et al.,

2021). Despite extensive research on curriculum design for the RL setting, existing techniques typically have limited theoretical underpinnings or require domain-specific hyperparameter tuning.

In this paper, we are interested in developing a principled curriculum strategy for the RL setting that is broadly applicable to many domains with minimal tuning of hyperparameters. To this end, we rely on the *Zone of Proximal Development* (ZPD) concept from the educational psychology literature (Vygotsky & Cole, 1978; Chaiklin, 2003). The ZPD concept, when applied in terms of learning progress, suggests that progress is maximized when the learner is presented with tasks that lie in the *proximal zone*, i.e., tasks that are neither too hard nor too easy. This idea of proximal zone can be captured using a notion of probability of success score $\text{PoS}_{\pi_t}(s)$ w.r.t. the learner's current policy $\pi_t$ for any given task $s$. Building on this idea, we mathematically derive an intuitive curriculum strategy by analyzing two simple learning settings. Our main results and contributions are as follows:

   I. We propose a curriculum strategy, PROCURL, inspired by the ZPD concept. PROCURL formalizes the idea of picking tasks that are neither too hard nor too easy for the learner in the form of selection strategy $\arg\max_s \text{PoS}_{\pi_t}(s) \cdot \left(\text{PoS}^*(s) - \text{PoS}_{\pi_t}(s)\right)$, where $\text{PoS}^*(s)$ corresponds to the probability of success score w.r.t. an optimal policy (Section 3.1).

  II. We derive PROCURL under two specific learning settings where we analyze the effect of picking a task on the agent's learning progress (Section 3.2).

 III. We present a practical variant of PROCURL, namely PROCURL-VAL, that can be easily integrated with deep RL frameworks with minimal hyperparameter tuning (Section 3.3).

  IV. We empirically demonstrate the effectiveness of PROCURL-VAL over state-of-the-art baselines in accelerating the training process of deep RL agents in a variety of environments (Section 4).[1]

## 1.1   Related Work

**Curriculum strategies based on domain knowledge.**   Early works on curriculum design for the supervised learning setting typically order the training examples in increasing difficulty (Elman, 1993; Bengio et al., 2009; Schmidhuber, 2013; Zaremba & Sutskever, 2014). This easy-to-hard design principle has been utilized in the hand-crafted curriculum approaches for the RL setting (Asada et al., 1996; Wu & Tian, 2016). Moreover, there have been recent works on designing greedy curriculum strategies for the imitation learning setting based on the iterative machine teaching framework (Liu et al., 2017; Yang et al., 2018; Zhu et al., 2018; Kamalaruban et al., 2019; Yengera et al., 2021). However, these approaches require domain-specific expert knowledge for designing difficulty measures.

**Curriculum strategies based on ZPD concept.**   In the pedagogical setting, it has been realized that effective teaching provides tasks that are neither too hard nor too easy for the human learner. This intuition of providing tasks from a particular range of difficulties is conceptualized in the ZPD concept (Vygotsky & Cole, 1978; Chaiklin, 2003; Oudeyer et al., 2007; Baranes & Oudeyer, 2013; Zou et al., 2019). In the RL setting, several curriculum strategies that have been proposed are inherently based on the ZPD concept (Florensa et al., 2017; 2018; Wöhlke et al., 2020). A common underlying theme in both Florensa et al. (2017) and Florensa et al. (2018) is that they choose the next task (starting or goal state) for the learner uniformly at random from the set $\{s : r_{\min} \leq \text{PoS}_{\pi_t}(s) \leq r_{\max}\}$. Here, the threshold values $r_{\min}$ and $r_{\max}$ require tuning according to the learner's progress and specific to the domain. Wöhlke et al. (2020) propose a unified framework for the learner's performance-based starting state curricula in RL. In particular, the starting state selection policy of Wöhlke et al. (2020), $\mathbb{P}\left[s_t^{(0)} = s\right] \propto G(\text{PoS}_{\pi_t}(s))$ for some function $G$, accommodates existing curriculum generation methods like Florensa et al. (2017); Graves et al. (2017). Despite promising empirical results, theoretical analysis of the impact of the chosen curriculum on the RL agent's learning progress is still missing in the aforementioned works.

**Curriculum strategies based on self-paced learning (SPL).** In the supervised learning setting, the curriculum strategies using the SPL concept optimize the trade-off between exposing the learner to all

---

[1]Github repo: `https://github.com/machine-teaching-group/tmlr2023_proximal-curriculum-rl`.

available training examples and selecting examples in which it currently performs well (Kumar et al., 2010; Jiang et al., 2015). In SPDL (Klink et al., 2020b;a; 2021; 2022) and SPaCE (Eimer et al., 2021), the authors have adapted the concept of SPL to the RL setting by controlling the intermediate task distribution with respect to the learner's current training progress. However, SPDL and SPaCE differ in their mode of operation and their objective. SPDL considers the procedural task generation framework where tasks of appropriate difficult levels can be synthesized, as also considered in Florensa et al. (2017; 2018). In contrast, SPaCE considers a pool-based curriculum framework for picking suitable tasks, as popular in the supervised learning setting. Further, SPDL considers the objective of a targeted performance w.r.t. a target distribution (e.g., concentrated distribution on hard tasks); in contrast, SPaCE considers the objective of uniform performance across a given pool of tasks. Similar to SPaCE, in our work, we consider the pool-based setting with uniform performance objective. Both SPDL and SPaCE serve as state-of-the-art baselines in our experimental evaluation. In terms of curriculum strategy, SPDL operates by solving an optimization problem at each step to pick a task (Klink et al., 2021); SPaCE uses a ranking induced by the magnitude of differences in current/previous critic values at each step to pick a task (Eimer et al., 2021). In the appendix, we have also provided some additional information on hyperparameters for SPDL and SPaCE.

**Other automatic curriculum strategies.** There are other approaches for automatic curriculum generation, including: (i) by formulating the curriculum design problem with the use of a meta-level Markov Decision Process (Narvekar et al., 2017; Narvekar & Stone, 2019); (ii) by learning how to generate training tasks similar to a teacher (Dendorfer et al., 2020; Such et al., 2020; Matiisen et al., 2019; Turchetta et al., 2020); (iii) by leveraging self-play as a form of curriculum generation (Sukhbaatar et al., 2018); (iv) by using the disagreement between different agents trained on the same tasks (Zhang et al., 2020); (v) by picking the starting states based on a single demonstration (Salimans & Chen, 2018; Resnick et al., 2018); and (vi) by providing agents with environment variations that are at the frontier of an agent's capabilities, e.g., Unsupervised Environment Design methods (Dennis et al., 2020; Jiang et al., 2021a; Parker-Holder et al., 2022). We refer the reader to recent surveys on curriculum design for the RL setting (Narvekar et al., 2020; Portelas et al., 2021; Weng, 2020).

## 2 Formal Setup

In this section, we formalize our problem setting based on prior work on teacher-student curriculum learning (Matiisen et al., 2019).

**MDP environment.** We consider a learning environment defined as a Markov Decision Process (MDP) $\mathcal{M} := (\mathcal{S}, \mathcal{A}, \mathcal{T}, H, R, \mathcal{S}_{\text{init}})$. Here, $\mathcal{S}$ and $\mathcal{A}$ denote the state and action spaces, $\mathcal{T} : \mathcal{S} \times \mathcal{S} \times \mathcal{A} \to [0, 1]$ is the transition dynamics, $H$ is the maximum length of the episode, and $R : \mathcal{S} \times \mathcal{A} \to \mathbb{R}$ is the reward function. The set of initial states $\mathcal{S}_{\text{init}} \subseteq \mathcal{S}$ specifies a fixed pool of *tasks*, i.e., each starting state $s \in \mathcal{S}_{\text{init}}$ corresponds to a unique task. Note that the above environment formalism is quite general enough to cover many practical settings, including the contextual multi-task MDP setting (Hallak et al., 2015).[2]

**RL agent and training process.** We consider an RL agent acting in this environment via a policy $\pi : \mathcal{S} \times \mathcal{A} \to [0, 1]$ that is a mapping from a state to a probability distribution over actions.[3] Given a task with the corresponding starting state $s \in \mathcal{S}_{\text{init}}$, the agent attempts the task via a trajectory rollout obtained by executing its policy $\pi$ from $s$ in the MDP $\mathcal{M}$. The trajectory rollout is denoted as $\xi = \{(s^{(\tau)}, a^{(\tau)}, R(s^{(\tau)}, a^{(\tau)}))\}_{\tau=0,1,\ldots,h}$ with $s^{(0)} = s$ and for some $h \leq H$. The agent's performance on task $s$ is measured via the value function $V^\pi(s) := \mathbb{E}\left[\sum_{\tau=0}^{h} R(s^{(\tau)}, a^{(\tau)}) \big| \pi, \mathcal{M}, s^{(0)} = s\right]$. Then, the uniform performance of the agent over the pool of tasks $\mathcal{S}_{\text{init}}$ is given by $V^\pi := \mathbb{E}_{s \sim \text{Uniform}(\mathcal{S}_{\text{init}})}[V^\pi(s)]$. The training process of the agent involves an interaction between two components: a student component that is responsible for policy update and a teacher component that is responsible for task selection. The interaction happens in discrete steps, indexed by $t = 1, 2, \ldots$, and is formally described in Algorithm 1.

---

[2]In this setting, for a given set of contexts $\mathcal{C}$, the pool of tasks is given by $\{\mathcal{M}_c = (\overline{\mathcal{S}}, \mathcal{A}, \mathcal{T}_c, H, R_c, \overline{\mathcal{S}}_{\text{init}}) : c \in \mathcal{C}\}$. Our environment formalism (MDP $\mathcal{M}$) covers this setting as follows: $\mathcal{S} = \overline{\mathcal{S}} \times \mathcal{C}$; $\mathcal{S}_{\text{init}} = \overline{\mathcal{S}}_{\text{init}} \times \mathcal{C}$; $\mathcal{T}((\bar{s}', c)|(\bar{s}, c), a) = \mathcal{T}_c(\bar{s}'|\bar{s}, a)$ and $R((\bar{s}, c), a) = R_c(\bar{s}, a), \forall \bar{s}, \bar{s}' \in \overline{\mathcal{S}}, a \in \mathcal{A}, c \in \mathcal{C}$.

[3]For general finite-horizon MDPs, including the time step as part of the state is important. However, to avoid complicating the notation with additional indexing, we have assumed that the time step is implicitly included in the state.

Let $\pi_{\text{end}}$ denote the agent's final policy at the end of training. The *training objective* is to ensure that the uniform performance of the policy $\pi_{\text{end}}$ is $\epsilon$-near-optimal, i.e., $(\max_\pi V^\pi - V^{\pi_{\text{end}}}) \leq \epsilon$. In the following two paragraphs, we discuss the student and teacher components in detail.

**Student component.** We consider a parametric representation for the RL agent, whose current knowledge is parameterized by $\theta \in \Theta \subseteq \mathbb{R}^d$ and each parameter $\theta$ is mapped to a policy $\pi_\theta : \mathcal{S} \times \mathcal{A} \to [0, 1]$. At step $t$, the student component updates the knowledge parameter based on the following quantities: the current knowledge parameter $\theta_t$, the task picked by the teacher component, and the rollout $\xi_t = \{(s_t^{(\tau)}, a_t^{(\tau)}, R(s_t^{(\tau)}, a_t^{(\tau)}))\}_\tau$. Then, the updated knowledge parameter $\theta_{t+1}$ is mapped to the agent's policy given by $\pi_{t+1} := \pi_{\theta_{t+1}}$. As a concrete example, the knowledge parameter of the REINFORCE agent (Sutton et al., 1999) is updated as $\theta_{t+1} \leftarrow \theta_t + \eta_t \cdot \sum_{\tau=0}^{h-1} G_t^{(\tau)} \cdot g_t^{(\tau)}$, where $\eta_t$ is the learning rate, $G_t^{(\tau)} = \sum_{\tau'=\tau}^{h} R(s_t^{(\tau')}, a_t^{(\tau')})$, and $g_t^{(\tau)} = \left[\nabla_\theta \log \pi_\theta(a_t^{(\tau)} | s_t^{(\tau)})\right]_{\theta=\theta_t}$.

**Teacher component.** At step $t$, the teacher component picks a task with the corresponding starting state $s_t^{(0)}$ for the student component to attempt via a trajectory rollout (see line 3 in Algorithm 1). The sequence of tasks (curriculum) picked by the teacher component affects the performance improvement of the policy $\pi_t$. The main focus of this work is to develop a teacher component to achieve the training objective in both a computational and a sample-efficient manner.

---

**Algorithm 1** RL Agent Training as Interaction between Teacher-Student Components

---

1: **Input:** RL agent's initial policy $\pi_1$
2: **for** $t = 1, 2, \ldots$ **do**
3:     Teacher component picks a task with the corresponding starting state $s_t^{(0)}$.
4:     Student component attempts the task via a trajectory rollout $\xi_t$ using the policy $\pi_t$ from $s_t^{(0)}$.
5:     Student component updates the policy to $\pi_{t+1}$.
6: **Output:** RL agent's final policy $\pi_{\text{end}} \leftarrow \pi_{t+1}$.

---

## 3 Proximal Curriculum Strategy

In Section 3.1, we propose a curriculum strategy for the goal-based setting. In Section 3.2, we show that the proposed curriculum strategy can be mathematically derived by analyzing simple learning settings. In Section 3.3, we present our final curriculum strategy that is applicable in general settings.

### 3.1 Curriculum Strategy for the Goal-based Setting

Here, we introduce our curriculum strategy for the goal-based setting using the notion of probability of success scores.

**Goal-based setting.** In this setting, the reward function $R$ is goal-based, i.e., the agent gets a reward of 1 only at the goal states and 0 at other states; moreover, any action from a goal state also leads to termination. For any task with the corresponding starting state $s \in \mathcal{S}_{\text{init}}$, we say that the attempted rollout $\xi$ succeeds in the task if the final state of $\xi$ is a goal state. Formally, $\text{succ}(\xi; s)$ is an indicator function whose value is 1 when the rollout $\xi$ succeeds in task $s$, and 0 otherwise. Furthermore, for an agent with policy $\pi$, we have that $V^\pi(s) := \mathbb{E}\left[\text{succ}(\xi; s) \big| \pi, \mathcal{M}\right]$ is equal to the total probability of reaching a goal state by executing the policy $\pi$ starting from $s \in \mathcal{S}_{\text{init}}$.

**Probability of success.** We begin by assigning a probability of success score for any task with the corresponding starting state $s \in \mathcal{S}_{\text{init}}$ w.r.t. any parameterized policy $\pi_\theta$ in the MDP $\mathcal{M}$.

**Definition 1.** *For any given knowledge parameter $\theta \in \Theta$ and any starting state $s \in \mathcal{S}_{\text{init}}$, we define the probability of success score $\text{PoS}_\theta(s)$ as the probability of successfully solving the task $s$ by executing the policy $\pi_\theta$ in the MDP $\mathcal{M}$. For the goal-based setting, we have $\text{PoS}_\theta(s) = V^{\pi_\theta}(s)$.*

With the above definition, the probability of success score for any task $s \in \mathcal{S}_{\text{init}}$ w.r.t. the agent's current policy $\pi_t$ is given by $\text{PoS}_t(s) := \text{PoS}_{\theta_t}(s)$. Further, we define $\text{PoS}^*(s) := \max_{\theta \in \Theta} \text{PoS}_\theta(s)$.

**Curriculum strategy.** Based on the notion of probability of success scores that we defined above, we propose the following curriculum strategy:

$$s_t^{(0)} \;\leftarrow\; \underset{s \in \mathcal{S}_{\text{init}}}{\arg\max} \Big( \text{PoS}_t(s) \cdot \big( \text{PoS}^*(s) - \text{PoS}_t(s) \big) \Big), \tag{1}$$

i.e., at step $t$, the teacher component picks a task associated with the starting state $s_t^{(0)}$ according to Eq. 1. The term $\text{PoS}_t(s) \cdot (\text{PoS}^*(s) - \text{PoS}_t(s))$ can be interpreted as the geometric mean of two quantities: the learner's probability of solving the task and the expected regret of the learner on this task. In the following subsection, we show that the above curriculum strategy can be derived by considering simple learning settings, such as contextual bandit problems with REINFORCE agent; these derivations provide insights about the design of the curriculum strategy.

## 3.2 Theoretical Justifications for the Curriculum Strategy

To derive our curriculum strategy for the goal-based setting, we additionally consider *independent tasks* where any task $s_t^{(0)}$ picked from the pool $\mathcal{S}_{\text{init}}$ at step $t$ only affects the agent's knowledge component corresponding to that task. Further, we assume that there exists a knowledge parameter $\theta^* \in \Theta$ such that $\pi_{\theta^*} \in \arg\max_\pi V^\pi$, and $\pi_{\theta^*}$ is referred to as the target policy. Then, based on the work of Weinshall et al. (2018); Kamalaruban et al. (2019); Yengera et al. (2021), we investigate the effect of picking a task $s_t^{(0)}$ at step $t$ on the convergence of the agent's parameter $\theta_t$ towards the target parameter $\theta^*$. Under a smoothness condition on the value function of the form $|V^{\pi_\theta} - V^{\pi_{\theta'}}| \leq L \cdot \|\theta - \theta'\|_1, \forall \theta, \theta' \in \Theta$ for some $L > 0$, we can translate the parameter convergence ($\theta_t \to \theta^*$) into the performance convergence ($V^{\pi_{\theta_t}} \to V^{\pi_{\theta^*}}$). Thus, we define the improvement in the training objective at step $t$ as

$$\Delta_t(\theta_{t+1}|\theta_t, s_t^{(0)}, \xi_t) \;:=\; [\|\theta^* - \theta_t\|_1 - \|\theta^* - \theta_{t+1}\|_1]. \tag{2}$$

In the above objective, we use the $\ell_1$-norm because our theoretical analysis considers the independent task setting mentioned above. Further, we define the expected improvement in the training objective at step $t$ due to picking the task $s_t^{(0)}$ as follows:

$$C_t(s_t^{(0)}) \;:=\; \mathbb{E}_{\xi_t|s_t^{(0)}} \big[ \Delta_t(\theta_{t+1}|\theta_t, s_t^{(0)}, \xi_t) \big]. \tag{3}$$

Note that the above quantity is an approximation of the expected learning progress measure as defined in Graves et al. (2017). In the following subsection, we justify our proposed curriculum strategy by analyzing the above quantity for a specific agent model under the independent task setting. More concretely, for the specific setting considered in Section 3.2.1, Theorem 1 implies that picking tasks based on the curriculum strategy given in Eq. 1 maximizes the expected value of the objective in Eq. 2. In the appendix, we provide an additional justification by considering an abstract agent model with a direct performance parameterization.

### 3.2.1 Reinforce Agent with Softmax Policy Parameterization

We consider the REINFORCE agent model with the following softmax policy parameterization: for any $\theta \in \mathbb{R}^{|\mathcal{S}| \cdot |\mathcal{A}|}$, we parameterize the policy as $\pi_\theta(a|s) \propto \exp(\theta[s, a]), \forall s \in \mathcal{S}, a \in \mathcal{A}$. In the following, we consider a problem instance involving a pool of contextual bandit tasks (a special case of independent task setting). Consider an MDP $\mathcal{M}$ with $g \in \mathcal{S}$ as the goal state for all tasks, $\mathcal{S}_{\text{init}} = \mathcal{S} \setminus \{g\}$, $\mathcal{A} = \{a_1, a_2\}$, and $H = 1$. We define the reward function as follows: $R(s, a) = 0, \forall s \in \mathcal{S} \setminus \{g\}, a \in \mathcal{A}$ and $R(g, a) = 1, \forall a \in \mathcal{A}$. For a given probability mapping $p_{\text{rand}} : \mathcal{S} \to [0, 1]$, we define the transition dynamics as follows: $\mathcal{T}(g|s, a_1) = p_{\text{rand}}(s), \forall s \in \mathcal{S}$; $\mathcal{T}(s|s, a_1) = 1 - p_{\text{rand}}(s), \forall s \in \mathcal{S}$; and $\mathcal{T}(s|s, a_2) = 1, \forall s \in \mathcal{S}$. Then, for the REINFORCE agent under the above setting, the following theorem quantifies the expected improvement in the training objective at step $t$:

**Theorem 1.** *Consider the REINFORCE agent with softmax policy parameterization under the independent task setting as described above. Let $s_t^{(0)}$ be the task picked at step $t$ with $\text{PoS}_{\theta_t}(s_t^{(0)}) = p$ and $\text{PoS}_{\theta^*}(s_t^{(0)}) = p^*$. Then, we have: $C_t(s_t^{(0)}) = 2 \cdot \eta_t \cdot p \cdot \left(1 - \frac{p}{p^*}\right)$, where $\eta_t$ is the learning of the REINFORCE agent.*

For the above setting with $p_{\text{rand}}(s) = 1, \forall s \in \mathcal{S}$, $\max_{s \in \mathcal{S}_{\text{init}}} C_t(s)$ is equivalent to $\max_{s \in \mathcal{S}_{\text{init}}} \text{PoS}_t(s) \cdot (1 - \text{PoS}_t(s))$. This means that for the case of $\text{PoS}^*(s) = 1, \forall s \in \mathcal{S}_{\text{init}}$, the curriculum strategy given in Eq. 1 can be seen as greedily optimizing the expected improvement in the training objective at step $t$ given in Eq. 3.

### 3.3 Curriculum Strategy for General Settings

Next, we discuss various practical issues in directly applying the curriculum strategy in Eq. 1 for general settings, and introduce several design choices to address these issues.

**Softmax selection.** When training deep RL agents, it is typically useful to allow some stochasticity in the selected batch of tasks. Moreover, the $\arg\max$ selection in Eq. 1 is brittle in the presence of any approximation errors in computing $\text{PoS}(\cdot)$ values. To tackle this issue, we replace $\arg\max$ selection in Eq. 1 with softmax selection and sample according to the following distribution:

$$\mathbb{P}\big[s_t^{(0)} = s\big] \ \propto \ \exp\Big(\beta \cdot \text{PoS}_t(s) \cdot \big(\text{PoS}^*(s) - \text{PoS}_t(s)\big)\Big), \tag{4}$$

where $\beta$ is a hyperparameter. Here, $\text{PoS}_t(s)$ values are computed for each $s \in \mathcal{S}_{\text{init}}$ using rollouts obtained via executing the policy $\pi_t$ in $\mathcal{M}$; $\text{PoS}^*(s)$ values are assumed to be provided as input.

$\text{PoS}^*(\cdot)$ **is not known.** Since the target policy $\pi_{\theta^*}$ is unknown, it is not possible to compute the $\text{PoS}^*(s)$ values without additional domain knowledge. In our experiments, we resort to simply setting $\text{PoS}^*(s) = 1, \forall s \in \mathcal{S}_{\text{init}}$ in Eq. 4 – the rationale behind this choice is that we expect the ideal $\pi_{\theta^*}$ to succeed in all the tasks in the pool.[4] This brings us to the following curriculum strategy referred to as PROCURL-ENV in our experimental evaluation:

$$\mathbb{P}\big[s_t^{(0)} = s\big] \ \propto \ \exp\Big(\beta \cdot \text{PoS}_t(s) \cdot \big(1 - \text{PoS}_t(s)\big)\Big). \tag{5}$$

**Computing** $\text{PoS}_t(\cdot)$ **is expensive.** It is expensive (sample inefficient) to estimate $\text{PoS}_t(s)$ over the space $\mathcal{S}_{\text{init}}$ using rollouts of the policy $\pi_t$. To tackle this issue, we replace $\text{PoS}_t(s)$ with values $V_t(s)$ obtained from the critic network of the RL agent. This brings us to the following curriculum strategy referred to as PROCURL-VAL in our experimental evaluation:

$$\mathbb{P}\big[s_t^{(0)} = s\big] \ \propto \ \exp\Big(\beta \cdot V_t(s) \cdot \big(1 - V_t(s)\big)\Big). \tag{6}$$

**Extension to non-binary or dense reward settings.** The current forms of PROCURL-VAL in Eq. 6 and PROCURL-ENV in Eq. 5 are not directly applicable for settings where the reward is non-binary or dense. To deal with this issue in PROCURL-VAL, we replace $V_t(s)$ values from the critic in Eq. 6 with normalized values given by $\overline{V}_t(s) = \frac{V_t(s) - V_{\min}}{V_{\max} - V_{\min}}$ clipped to the range $[0, 1]$. Here, $V_{\min}$ and $V_{\max}$ could be provided as input based on the environment's reward function; alternatively we can dynamically set $V_{\min}$ and $V_{\max}$ during the training process by taking min-max values of the critic for states $\mathcal{S}_{\text{init}}$ at step $t$. To deal with this issue in PROCURL-ENV, we replace $\text{PoS}_t(s)$ values from the rollouts in Eq. 5 with normalized values $\overline{V}_t(s)$ as above. Algorithm 2 in the appendix provides a complete pseudo-code for the RL agent training with PROCURL-VAL in this general setting.

## 4 Experimental Evaluation

In this section, we evaluate the effectiveness of our curriculum strategies on a variety of domains w.r.t. the uniform performance of the trained RL agent over the training pool of tasks. Additionally, we consider the following two metrics in our evaluation: (i) total number of environment steps incurred jointly by the teacher and the student components at the end of the training process; (ii) total clock time required for the training process. Throughout all the experiments, we use the PPO method from Stable-Baselines3 library for policy optimization (Schulman et al., 2017; Raffin et al., 2021).

---

[4]This simple choice leads to competitive performance in a variety of environments used in our experiments. However, the above choice could lead to a suboptimal strategy for specific scenarios, e.g., when all $\text{PoS}^*(s)$ are below 0.5. It would be interesting to investigate alternative strategies to estimate $\text{PoS}^*(s)$ during the training process, e.g., using top $K\%$ rollouts obtained by executing the current policy $\pi_t$ starting from $s$.

| Environment | Reward | Context | State | Action | Pool size |
|---|---|---|---|---|---|
| PointMass-s | binary | $\mathbb{R}^3$ | $\mathbb{R}^4$ | $\mathbb{R}^2$ | 100 |
| PointMass-d | non-binary | $\mathbb{R}^3$ | $\mathbb{R}^4$ | $\mathbb{R}^2$ | 100 |
| BasicKarel | binary | 24000 | $\{0,1\}^{88}$ | 6 | 24000 |
| BallCatching | non-binary | $\mathbb{R}^3$ | $\mathbb{R}^{21}$ | $\mathbb{R}^5$ | 100 |
| AntGoal | non-binary | $\mathbb{R}^2$ | $\mathbb{R}^{29}$ | $\mathbb{R}^8$ | 50 |

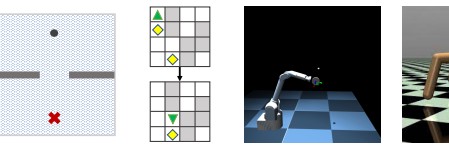 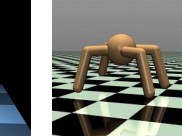

(a) Complexity of the environments  (b) Illustration of the environments

Figure 1: **(a)** shows complexity of the environments w.r.t. the reward signals, context variation, state space, action space, and the pool size of the tasks used for training. **(b)** shows illustration of the environments (from left to right): PointMass, BasicKarel, BallCatching, and AntGoal. Details are provided in Section 4.1.

## 4.1 Environments

We consider 5 different environments in our evaluation, as described in the following paragraphs. Figure 1 provides a summary and illustration of these environments.

**PointMass-s and PointMass-d.** Based on the work of Klink et al. (2020b), we consider a contextual PointMass environment where an agent navigates a point mass through a gate of a given size towards a goal in a two-dimensional space. More concretely, we consider two settings: (i) PointMass-s environment corresponds to a goal-based (i.e., binary and sparse) reward setting where the agent receives a reward of 1 only if it successfully moves the point mass to the goal position; (ii) PointMass-d environment corresponds to a dense reward setting as used by Klink et al. (2020b) where the reward values decay in a squared exponential manner with increasing distance to the goal. Here, the contextual variable $c \in \mathbb{R}^3$ controls the position of the gate (*C-GatePosition*), the width of the gate (*C-GateWidth*), and the friction coefficient of the ground (*C-Friction*). We construct the training pool of tasks by uniformly sampling 100 tasks over the space of possible tasks (here, each task corresponds to a different contextual variable).

**BasicKarel.** This environment is inspired by the Karel program synthesis domain Bunel et al. (2018), where the goal of an agent is to transform an initial grid into a final grid configuration by a sequence of commands. In our BasicKarel environment, we do not allow any programming constructs such as conditionals or loops and limit the commands to the "basic" actions given by $\mathcal{A} = \{\texttt{move}, \texttt{turnLeft}, \texttt{turnRight}, \texttt{pickMarker}, \texttt{putMarker}, \texttt{finish}\}$. A task in this environment corresponds to a pair of initial grid and final grid configurations; the environment is episodic with goal-based (i.e., binary and sparse) reward setting where the agent receives a reward of 1 only if it successfully transforms the task's initial grid into the task's final grid. Here, the contextual variable is discrete, where each task can be considered as a discrete context. We construct the training pool of tasks by sampling 24000 tasks; additional details are provided in the appendix.

**BallCatching.** This environment is the same used in the work of Klink et al. (2020b); here, an agent needs to direct a robot to catch a ball thrown towards it. The reward function is sparse and non-binary, only rewarding the robot when it catches the ball and penalizing it for excessive movements. The contextual vector $c \in \mathbb{R}^3$ captures the distance to the robot from which the ball is thrown and its goal position in a plane that intersects the base of the robot. We construct the training pool of tasks by uniformly sampling 100 tasks over the space of possible tasks.

**AntGoal.** This environment is adapted from the original MuJoCo Ant environment (Todorov et al., 2012). In our adaptation, we additionally have a goal on a flat 2D surface, and an agent is rewarded for moving an ant robot towards the goal location. This goal-based reward term replaces the original reward term of making the ant move forward; also, this reward term increases exponentially when the ant moves closer to the goal location. We keep the other reward terms, such as control and contact costs, similar to the original MuJoCo Ant environment. The environment is episodic with a length of 200 steps. The goal location essentially serves as a contextual variable in $\mathbb{R}^2$. We construct the training pool of tasks by uniformly sampling 50 goal locations from a circle around the ant.

These environments are goal-based and have an implicit way of defining a successful trajectory. Typically, success is defined as a reward signal to the agent for approaching the goal, as done by Klink et al. (2020b) for POINTMASS, BALLCATCHING, and ANTGOAL. As future work, it would also be interesting to investigate the effect of our curriculum strategy on RL algorithms designed for the same goal-based setting but without assuming that a goal proximity function is defined in the environment (Ding et al., 2019; Eysenbach et al., 2022; Lin et al., 2019).

## 4.2 Curriculum Strategies Evaluated

**Variants of our curriculum strategy.** We consider the curriculum strategies PROCURL-VAL and PROCURL-ENV from Section 3.3. Since PROCURL-ENV uses policy rollouts to estimate $\mathrm{PoS}_t(s)$ in Eq. 5, it requires environment steps for selecting tasks in addition to environment steps for training. To compare PROCURL-VAL and PROCURL-ENV in terms of trade-off between performance and sample efficiency, we introduce a variant PROCURL-ENV$^X$ where x controls the budget of the total number of steps used for estimation and training. In Figure 3, variants with $x \in \{2, 4\}$ refer to a total budget of about x million environment steps when training comprises of 1 million steps.

**State-of-the-art baselines.** SPDL (Klink et al., 2020b) and SPACE (Eimer et al., 2021) are state-of-the-art curriculum strategies for contextual RL. We adapt the implementation of an improved version of SPDL, presented in Klink et al. (2021), to work with a discrete pool of tasks. We also introduce a variant of SPACE, namely SPACE-ALT, by adapting the implementation of Eimer et al. (2021) to sample the next training task as $\mathbb{P}\big[s_t^{(0)} = s\big] \propto \exp\big(\beta \cdot \big(V_t(s) - V_{t-1}(s)\big)\big)$. PLR (Jiang et al., 2021b) is a state-of-the-art curriculum strategy originally designed for procedurally generated content settings. We adapt the implementation of PLR for the contextual RL setting operating on a fixed pool of tasks and include it as an additional baseline.

**Prototypical baselines.** IID strategy randomly samples the next task from the pool; note that IID serves as a competitive baseline since we consider the uniform performance objective. We introduce two additional variants of PROCURL-ENV, namely EASY and HARD, to understand the importance of the two terms $\mathrm{PoS}_t(s)$ and $\big(1 - \mathrm{PoS}_t(s)\big)$ in Eq. 5. EASY samples tasks as $\mathbb{P}\big[s_t^{(0)} = s\big] \propto \exp\big(\beta \cdot \mathrm{PoS}_t(s)\big)$, and HARD samples tasks as $\mathbb{P}\big[s_t^{(0)} = s\big] \propto \exp\big(\beta \cdot \big(1 - \mathrm{PoS}_t(s)\big)\big)$.

## 4.3 Results

**Convergence behavior and curriculum plots.** As shown in Figure 2, the RL agents trained using the variants of our curriculum strategy, PROCURL-ENV and PROCURL-VAL, either match or outperform the agents trained with state-of-the-art and prototypical baselines in all the environments. Figures 4 and 5 visualize the curriculums generated by PROCURL-ENV, PROCURL-VAL, and IID; the trends for PROCURL-VAL generally indicate a gradual shift towards harder tasks across different contexts. The increasing trend in Figure 4a corresponds to a preference shift towards tasks with the gate positioned closer to the edges; the decreasing trend in Figure 4b corresponds to a preference shift towards tasks with narrower gates. For BASICKAREL, the increasing trends in Figures 5a and 5b correspond to a preference towards tasks with longer solution trajectories and tasks requiring a marker to be picked or put, respectively. In Figures 5c and 5d, tasks with a distractor marker (*C-DistractorMarker*) and tasks with more walls (*C-Walls*) are increasingly selected while training.

**Metrics comparison.** In Figure 3, we compare curriculum strategies considered in our experiments w.r.t. different metrics. PROCURL-VAL has similar sample complexity as state-of-the-art baselines since it does not require additional environment steps for the teacher component. PROCURL-VAL performs better compared to SPDL, SPACE and PLR in terms of computational complexity. The effect of that is more evident as the pool size increases. The reason is that PROCURL-VAL only requires forward-pass operation on the critic-model to obtain value estimates for each task in the pool. SPDL and SPACE not only require the same forward-pass operations, but SPDL does an additional optimization step, and SPACE requires a task ordering step. As for PLR, it has an additional computational overhead for scoring the sampled tasks. In terms of agent's performance, our curriculum strategies exceed or match these baselines at different training segments. Even though PROCURL-ENV consistently surpasses all the other variants in terms of perfor-

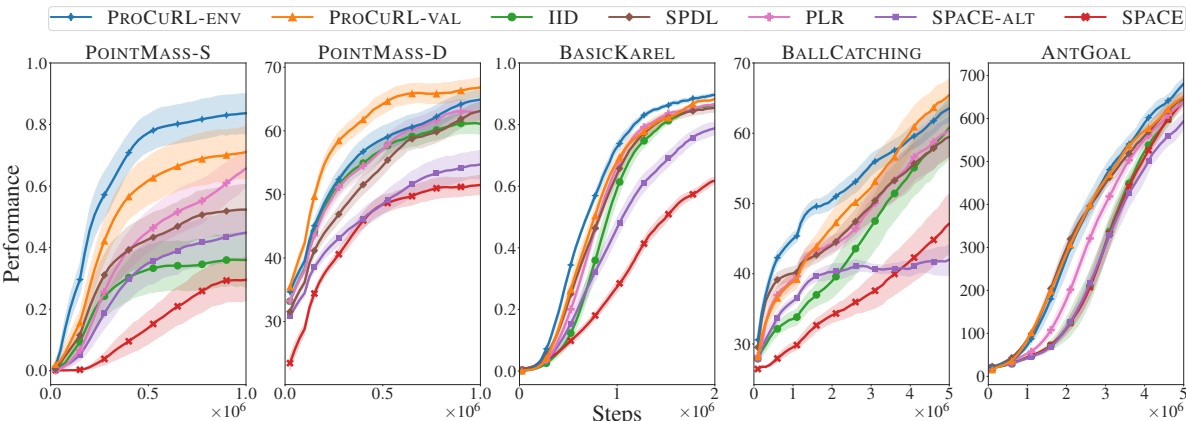

Figure 2: Performance comparison of RL agents trained using different curriculum strategies described in Section 4.2. The performance is measured as the mean reward ($\pm 1$ standard error) on the training pool of tasks. The results are averaged over 20 random seeds for POINTMASS-S and POINTMASS-D, 10 random seeds for BASICKAREL and BALLCATCHING, and 5 random seeds for ANTGOAL. The plots are smoothed across 5 evaluation snapshots happening at over 25000 training steps.

| Env / Method | POINTMASS-S | | | | | BASICKAREL | | | | |
|---|---|---|---|---|---|---|---|---|---|---|
| | Performance | | | Steps | Time | Performance | | | Steps | Time |
| | 0.25M | 0.5M | 1M | 1M | 1M | 0.25M | 0.5M | 1M | 1M | 1M |
| PROCURL-ENV | $0.60 \pm 0.16$ | $0.79 \pm 0.13$ | $0.84 \pm 0.14$ | $17.4 \pm 1.7$ | 156 | $0.10 \pm 0.02$ | $0.38 \pm 0.03$ | $0.76 \pm 0.04$ | $34.2 \pm 0.9$ | 191 |
| PROCURL-ENV[4] | $0.50 \pm 0.15$ | $0.64 \pm 0.15$ | $0.71 \pm 0.15$ | $4.0 \pm 0.0$ | 43 | $0.10 \pm 0.03$ | $0.38 \pm 0.04$ | $0.75 \pm 0.04$ | $4.6 \pm 0.1$ | 53 |
| PROCURL-ENV[2] | $0.36 \pm 0.17$ | $0.53 \pm 0.16$ | $0.60 \pm 0.17$ | $2.0 \pm 0.0$ | 25 | $0.10 \pm 0.03$ | $0.32 \pm 0.05$ | $0.73 \pm 0.04$ | $2.4 \pm 0.1$ | 44 |
| PROCURL-VAL | $0.48 \pm 0.15$ | $0.64 \pm 0.17$ | $0.71 \pm 0.18$ | $1.0 \pm 0.0$ | 20 | $0.06 \pm 0.03$ | $0.30 \pm 0.08$ | $0.71 \pm 0.05$ | $1.0 \pm 0.0$ | 70 |
| SPACE | $0.05 \pm 0.06$ | $0.17 \pm 0.12$ | $0.29 \pm 0.15$ | $1.0 \pm 0.0$ | 22 | $0.04 \pm 0.02$ | $0.11 \pm 0.03$ | $0.30 \pm 0.04$ | $1.0 \pm 0.0$ | 89 |
| SPACE-ALT | $0.22 \pm 0.12$ | $0.37 \pm 0.15$ | $0.46 \pm 0.17$ | $1.0 \pm 0.0$ | 21 | $0.05 \pm 0.03$ | $0.18 \pm 0.06$ | $0.50 \pm 0.08$ | $1.0 \pm 0.0$ | 69 |
| SPDL | $0.34 \pm 0.16$ | $0.45 \pm 0.17$ | $0.52 \pm 0.17$ | $1.0 \pm 0.0$ | 23 | $0.07 \pm 0.02$ | $0.29 \pm 0.04$ | $0.69 \pm 0.05$ | $1.0 \pm 0.0$ | 81 |
| PLR | $0.32 \pm 0.12$ | $0.47 \pm 0.15$ | $0.69 \pm 0.13$ | $1.0 \pm 0.0$ | 20 | $0.05 \pm 0.03$ | $0.23 \pm 0.05$ | $0.70 \pm 0.04$ | $1.0 \pm 0.0$ | 79 |
| IID | $0.27 \pm 0.15$ | $0.34 \pm 0.17$ | $0.36 \pm 0.19$ | $1.0 \pm 0.0$ | 20 | $0.03 \pm 0.02$ | $0.15 \pm 0.06$ | $0.64 \pm 0.08$ | $1.0 \pm 0.0$ | 34 |
| EASY | $0.37 \pm 0.13$ | $0.44 \pm 0.12$ | $0.50 \pm 0.11$ | $17.1 \pm 2.3$ | 154 | $0.04 \pm 0.01$ | $0.07 \pm 0.02$ | $0.11 \pm 0.03$ | $22.6 \pm 0.9$ | 126 |
| HARD | $0.01 \pm 0.01$ | $0.00 \pm 0.00$ | $0.01 \pm 0.01$ | $37.0 \pm 0.7$ | 332 | $0.01 \pm 0.00$ | $0.01 \pm 0.00$ | $0.01 \pm 0.00$ | $35.2 \pm 3.9$ | 197 |

Figure 3: Comparison of different curriculum strategies described in Section 4.2 under the following metrics: (i) performance (mean reward $\pm$ $t \times$standard error, where $t$ is the value from the t-distribution table for 95% confidence (Beyer, 2019)) of the RL agent at 0.25, 0.5, and 1 million training steps; (ii) total number of environment steps incurred at the end of 1 million training steps (this captures the sample efficiency of a curriculum strategy); (iii) total clock time in minutes at the end of 1 million training steps (this captures the computational efficiency of a curriculum strategy).

mance, its teacher component requires a lot of additional environment steps. Regarding the prototypical baselines in Figure 3, we make the following observations: (a) IID is a strong baseline in terms of sample and computational efficiency; however, its performance tends to be unstable in POINTMASS-S environment because of high randomness; (b) EASY performs well in POINTMASS-S because of the presence of easy tasks in the task space of this environment, but, performs quite poorly in BASICKAREL; (c) HARD consistently fails in both the environments.

**Ablation and robustness experiments.** We conduct additional experiments to evaluate the robustness of PROCURL-VAL w.r.t. different values of $\beta$ and different $\epsilon$-level noise in $V_t(s)$ values. The results are reported in the appendix. From the reported results, we note that picking a value for $\beta$ somewhere between 10 to 30

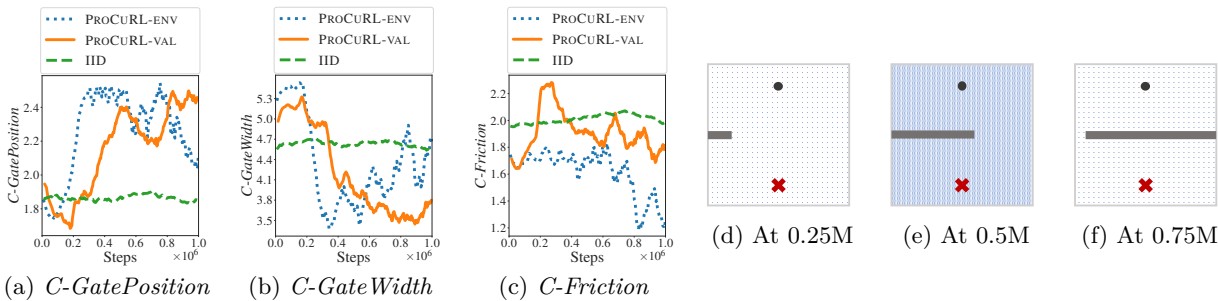

Figure 4: **(a-c)** Curriculum visualization of PROCURL-ENV, PROCURL-VAL, and IID in the POINTMASS-S environment; these plots show the moving average variation of the context variables of every 100 tasks picked by curriculum strategies during the training process (a picked task involves multiple training steps shown on the x-axis of plots). The increasing trend in **(a)** corresponds to a preference shift towards tasks with the gate positioned closer to the edges; the decreasing trend in **(b)** corresponds to a shift towards tasks with narrower gates. **(d-f)** Illustrative tasks used during the training process for PROCURL-VAL (M is $10^6$).

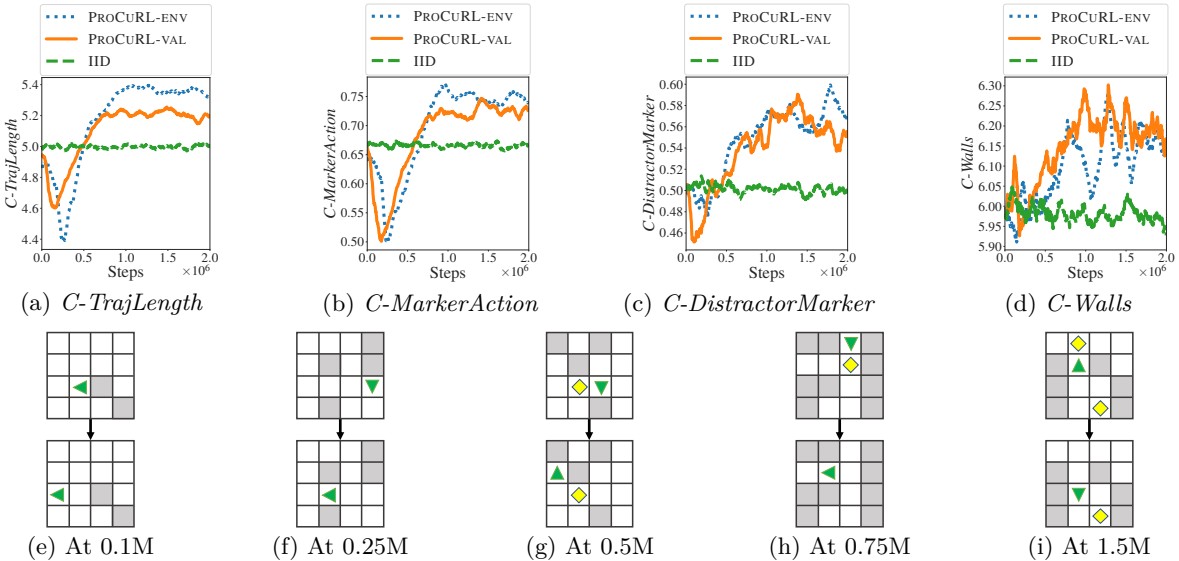

Figure 5: **(a-d)** Curriculum visualization of PROCURL-ENV, PROCURL-VAL, and IID in the BASICKAREL environment; these plots show the moving average variation of the context variables of every 500 tasks picked. The increasing trends in **(a-d)** correspond to a preference towards tasks: **(a)** with longer trajectories, **(b)** requiring a marker action, **(c)** with more distractor markers, and **(d)** with more walls. **(e-i)** Illustrative tasks used during the training process at different steps for PROCURL-VAL (M is $10^6$).

leads to competitive performance, and PROCURL-VAL is robust even for noise levels up to $\epsilon = 0.2$. Further, we conduct an ablation study on the form of our curriculum objective presented in Eq. 1. More specifically, we consider the following generalized variant of Eq. 1 with parameters $\gamma_1$ and $\gamma_2$: $s_t^{(0)} \leftarrow \arg\max_{s \in \mathcal{S}_{\text{init}}} \left( \text{PoS}_t(s) \cdot (\gamma_1 \cdot \text{PoS}^*(s) - \gamma_2 \cdot \text{PoS}_t(s)) \right)$. In our experiments, we consider the following range of $\gamma_2/\gamma_1 \in \{0.6, 0.8, 1.0, 1.2, 1.4\}$. The results are reported in the appendix. From the reported results, we note that our default curriculum strategy in Eq. 1 (corresponding to $\gamma_2/\gamma_1 = 1.0$) leads to competitive performance.

## 5  Concluding Discussions

We proposed a novel curriculum strategy for deep RL agents inspired by the ZPD concept. We mathematically derived our strategy by analyzing simple learning settings and empirically demonstrated its effectiveness

in a variety of complex domains. Here, we discuss a few limitations of our work and outline a plan on how to address them in future work. First, our experimental results show that different variants of our proposed curriculum provide an inherent trade-off between runtime and performance; it would be interesting to systematically study these variants to obtain a more effective curriculum strategy across different metrics. Second, it would be interesting to extend our curriculum strategy to sparse reward environments with high-dimensional context space; in particular, our curriculum strategy requires estimating the probability of success of all tasks in the pool when sampling a new task which becomes challenging in these environments. Third, extending the theoretical analysis of the curriculum strategy from independent task settings to correlated task settings would be an interesting avenue to explore; this could involve developing a generalized version of ProCuRL curriculum strategy using a distance metric over the context space (Klink et al., 2022; Huang et al., 2022).

## Acknowledgments

Parameswaran Kamalaruban acknowledges support from The Alan Turing Institute. Funded/Co-funded by the European Union (ERC, TOPS, 101039090). Views and opinions expressed are however those of the author(s) only and do not necessarily reflect those of the European Union or the European Research Council. Neither the European Union nor the granting authority can be held responsible for them.

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

## A    Table of Contents

In this section, we give a brief description of the content provided in the appendices of the paper.

- Appendix B provides a proof for Theorem 1 and an additional theoretical justification for our curriculum strategy. (Section 3.2)

- Appendix C provides additional details and results for experimental evaluation. (Section 4)

## B    Theoretical Justifications for the Curriculum Strategy – Proof and Additional Justification (Section 3.2)

### B.1    Proof of Theorem 1

*Proof.* For the contextual bandit setting described in Section 3.2.1, the REINFORCE learner's update rule reduces to the following: $\theta_{t+1} \leftarrow \theta_t + \eta_t \cdot \mathbf{1}\{s_t^{(1)} = g\} \cdot \left[\nabla_\theta \log \pi_\theta(a_t^{(0)}|s_t^{(0)})\right]_{\theta=\theta_t}$. In particular, for $s_t^{(0)} = s$ and $a_t^{(0)} = a_1$, we update:

$$\theta_{t+1}[s, a_1] \leftarrow \theta_t[s, a_1] + \eta_t \cdot \mathbf{1}\{s_t^{(1)} = g\} \cdot (1 - \pi_{\theta_t}(a_1|s))$$
$$\theta_{t+1}[s, a_2] \leftarrow \theta_t[s, a_2] - \eta_t \cdot \mathbf{1}\{s_t^{(1)} = g\} \cdot (1 - \pi_{\theta_t}(a_1|s))$$

and we set $\theta_{t+1}[s, \cdot] \leftarrow \theta_t[s, \cdot]$ when $s_t^{(0)} \neq s$ or $a_t^{(0)} \neq a_1$. Let $s_t^{(0)} = s$, and consider the following:

$$
\begin{aligned}
&\Delta_t(\theta_{t+1}|\theta_t, s, \xi_t) \\
=~& \|\theta^* - \theta_t\|_1 - \|\theta^* - \theta_{t+1}\|_1 \\
=~& \|\theta^*[s, \cdot] - \theta_t[s, \cdot]\|_1 - \|\theta^*[s, \cdot] - \theta_{t+1}[s, \cdot]\|_1 \\
=~& \{\theta^*[s, a_1] - \theta_t[s, a_1] + \theta_t[s, a_2] - \theta^*[s, a_2]\} - \{\theta^*[s, a_1] - \theta_{t+1}[s, a_1] + \theta_{t+1}[s, a_2] - \theta^*[s, a_2]\} \\
=~& \theta_{t+1}[s, a_1] - \theta_t[s, a_1] + \theta_t[s, a_2] - \theta_{t+1}[s, a_2] \\
=~& 2 \cdot \eta_t \cdot \mathbf{1}\{a_t^{(0)} = a_1, s_t^{(1)} = g\} \cdot (1 - \pi_{\theta_t}(a_1|s)).
\end{aligned}
$$

For the contextual bandit setting, the probability of success is given by $\mathrm{PoS}_\theta(s) = V^{\pi_\theta}(s) = p_{\mathrm{rand}}(s) \cdot \pi_\theta(a_1|s), \forall s \in \mathcal{S}$. We assume that $\exists~\theta^*$ such that $\pi_{\theta^*}(a_1|s) \rightarrow 1$; here, $\pi_{\theta^*}$ is the target policy. With the above definition, the probability of success scores for any task associated with the starting state $s \in \mathcal{S}_{\mathrm{init}}$ w.r.t. the target and agent's current policies (at any step $t$) are respectively given by $\mathrm{PoS}^*(s) = \mathrm{PoS}_{\theta^*}(s) = p_{\mathrm{rand}}(s) = p^*$ and $\mathrm{PoS}_t(s) = \mathrm{PoS}_{\theta_t}(s) = p_{\mathrm{rand}}(s) \cdot \pi_{\theta_t}(a_1|s) = p$. Now, we consider the following:

$$
\begin{aligned}
C_t(s) &= \mathbb{E}_{\xi_t|s}\left[\Delta_t(\theta_{t+1}|\theta_t, s, \xi_t)\right] \\
&= \mathbb{E}_{\xi_t|s}\left[2 \cdot \eta_t \cdot \mathbf{1}\{a_t^{(0)} = a_1, s_t^{(1)} = g\} \cdot (1 - \pi_{\theta_t}(a_1|s))\right] \\
&= 2 \cdot \eta_t \cdot p_{\mathrm{rand}}(s) \cdot \pi_{\theta_t}(a_1|s) \cdot (1 - \pi_{\theta_t}(a_1|s)) \\
&= 2 \cdot \eta_t \cdot p \cdot \left(1 - \frac{p}{p^*}\right).
\end{aligned}
$$

$\square$

### B.2    Abstract Agent with Direct performance Parameterization

We consider an abstract agent model with the following direct performance parameterization: for any $\theta \in \Theta = [0, 1]^{|\mathcal{S}_{\mathrm{init}}|}$, we have $\mathrm{PoS}_\theta(s) = \theta[s], \forall s \in \mathcal{S}_{\mathrm{init}}$.[5] Under this model, the agent's current knowledge $\theta_t$ at step $t$ is

---

[5] In this setting, we abstract out the policy $\pi_\theta$ and directly map the "parameter" $\theta$ to a vector of "performance on tasks" $\mathrm{PoS}_\theta$. Then, we choose the parameter space as $\Theta = [0, 1]^{\mathcal{S}_{\mathrm{init}}}$ (where $d = \mathcal{S}_{\mathrm{init}}$) and define $\mathrm{PoS}_\theta = \theta$. Thus, an update in the "parameter" $\theta$ is equivalent to an update in the "performance on tasks" $\mathrm{PoS}_\theta$.

encoded directly by its probability of success scores $\{\text{PoS}_{\theta_t}(s) \mid s \in \mathcal{S}_{\text{init}}\}$. The target knowledge parameter $\theta^*$ is given by $\{\text{PoS}_{\theta^*}(s) \mid s \in \mathcal{S}_{\text{init}}\}$. Under the independent task setting, we design an update rule for the agent to reflect the characteristics of the policy gradient style update. In particular, for $s = s_t^{(0)} \in \mathcal{S}_{\text{init}}$, we update

$$\theta_{t+1}[s] \; \leftarrow \; \theta_t[s] + \alpha \cdot \text{succ}(\xi_t; s) \cdot (\theta^*[s] - \theta_t[s]) + \beta \cdot (1 - \text{succ}(\xi_t; s)) \cdot (\theta^*[s] - \theta_t[s]),$$

where $\alpha, \beta \in [0, 1]$ and $\alpha > \beta$. For $s \in \mathcal{S}_{\text{init}}$ and $s \neq s_t^{(0)}$, we maintain $\theta_{t+1}[s] \leftarrow \theta_t[s]$. Importantly, $\alpha > \beta$ implies that the agent's current knowledge for the picked task is updated more when the agent succeeds in that task compared to the failure case. The update rule captures the following idea: when picking a task that is "too easy", the progress in $\theta_t$ towards $\theta^*$ is minimal since $(\theta^*[s] - \theta_t[s])$ is low; similarly, when picking a task that is "too hard", the progress in $\theta_t$ towards $\theta^*$ is minimal since $\beta \cdot (\theta^*[s] - \theta_t[s])$ is low for $\beta \ll 1$. This idea aligns with the ZPD concept in terms of the learning progress (Vygotsky & Cole, 1978; Chaiklin, 2003). For the abstract agent under the above setting, the following theorem quantifies the expected improvement in the training objective at step $t$:

**Theorem 2.** *Consider the abstract agent with direct performance parameterization under the independent task setting as described above. Let $s_t^{(0)}$ be the task picked at step $t$ with $\text{PoS}_{\theta_t}(s_t^{(0)}) = p$ and $\text{PoS}_{\theta^*}(s_t^{(0)}) = p^*$. Then, we have: $C_t(s_t^{(0)}) = \alpha \cdot p \cdot (p^* - p) + \beta \cdot (1 - p) \cdot (p^* - p)$.*

*Proof.* Let $s_t^{(0)} = s \in \mathcal{S}_{\text{init}}$, and consider the following:

$$\begin{aligned}
\Delta_t(\theta_{t+1} | \theta_t, s, \xi_t) &= \|\theta^* - \theta_t\|_1 - \|\theta^* - \theta_{t+1}\|_1 \\
&= \theta_{t+1}[s] - \theta_t[s] \\
&= \alpha \cdot \text{succ}(\xi_t; s) \cdot (\theta^*[s] - \theta_t[s]) + \beta \cdot (1 - \text{succ}(\xi_t; s)) \cdot (\theta^*[s] - \theta_t[s]).
\end{aligned}$$

For the abstract learner model defined in Section B.2, we have $\text{PoS}_\theta(s) = V^{\pi_\theta}(s) = \theta[s]$, for any $s \in \mathcal{S}_{\text{init}}$. Then, the probability of success scores for any task $s \in \mathcal{S}_{\text{init}}$ w.r.t. the target and agent's current policies (at any step $t$) are respectively given by $\text{PoS}^*(s) = \text{PoS}_{\theta^*}(s) = \theta^*[s] = p^*$ and $\text{PoS}_t(s) = \text{PoS}_{\theta_t}(s) = \theta_t[s] = p$. Now, we consider the following:

$$\begin{aligned}
C_t(s) &= \mathbb{E}_{\xi_t|s} \left[ \Delta_t(\theta_{t+1} | \theta_t, s, \xi_t) \right] \\
&= \mathbb{E}_{\xi_t|s} \left[ \alpha \cdot \text{succ}(\xi_t; s) \cdot (\theta^*[s] - \theta_t[s]) + \beta \cdot (1 - \text{succ}(\xi_t; s)) \cdot (\theta^*[s] - \theta_t[s]) \right] \\
&= \alpha \cdot \theta_t[s] \cdot (\theta^*[s] - \theta_t[s]) + \beta \cdot (1 - \theta_t[s]) \cdot (\theta^*[s] - \theta_t[s]) \\
&= \alpha \cdot p \cdot (p^* - p) + \beta \cdot (1 - p) \cdot (p^* - p).
\end{aligned}$$

$\square$

For the above setting with $\alpha = 1$ and $\beta = 0$, $\max_{s \in \mathcal{S}_{\text{init}}} C_t(s)$ is equivalent to $\max_{s \in \mathcal{S}_{\text{init}}} \text{PoS}_t(s) \cdot (\text{PoS}^*(s) - \text{PoS}_t(s))$. This, in turn, implies that the curriculum strategy given in Eq. 1 can be seen as greedily optimizing the expected improvement in the training objective at step $t$ given in Eq. 3.

## C   Experimental Evaluation – Additional Details (Section 4)

### C.1   Environments

**BasicKarel.** This environment is inspired by the Karel program synthesis domain Bunel et al. (2018), where the goal of an agent is to transform an initial grid into a final grid configuration by a sequence of commands. In the BASICKAREL environment, we do not allow any programming constructs such as conditionals or loops and limit the commands to the "basic" actions given by the action space $\mathcal{A} = \{\texttt{move}, \texttt{turnLeft}, \texttt{turnRight}, \texttt{pickMarker}, \texttt{putMarker}, \texttt{finish}\}$. A task in this environment corresponds to a pair of initial grid and final grid configurations. It consists of an avatar, walls, markers, and empty grid cells, and each element has a specific location in the grid. The avatar is characterized by its current location and orientation. Its orientation can be any direction $\{\texttt{North}, \texttt{East}, \texttt{South}, \texttt{West}\}$, and its location can be any grid cell, except from grid cells where a wall is located. The state space $\mathcal{S}$ of

BASICKAREL is any possible configuration of the avatar, walls, and markers in a pair of grids. The avatar can move around the grid and is directed via the basic Karel commands, i.e., the action space $\mathcal{A}$. While the avatar moves, if it hits a wall or the grid boundary, it "crashes" and the episode terminates. If `pickMarker` is selected when no marker is present, the avatar "crashes" and the program ends. Likewise, if the `putMarker` action is taken and a marker is already present, the avatar "crashes" and the program terminates. The `finish` action indicates the end of the sequence of actions, i.e., the episode ends after encountering this action. To successfully solve a BASICKAREL task, the sequence of actions must end with a `finish`, and there should be no termination via "crashes". Based on this environment, we created a multi-task dataset that consists of 24000 training tasks and 2400 test tasks. All the generated tasks have a grid size of $4 \times 4$.

## C.2 Evaluation Setup

**Hyperparameters of PPO method.** We use the PPO method from Stable-Baselines3 library with a basic MLP policy for all the conducted experiments (Schulman et al., 2017; Raffin et al., 2021). For the POINTMASS-S, POINTMASS-D, and BALLCATCHING environments, the MLP policy has a shared layer with 64 units and a second layer with separate 64 units for the policy and 64 units for the value function. For the BASICKAREL environment, we use two separate layers of size [512, 256] for the policy network and two layers of size [256, 128] for the value function network. For the ANTGOAL environment, we use two separate layers of size [512, 512] for the policy network and two layers of size [512, 512] for the value function network. For all the experiments, ReLU is the chosen activation function. In Figure 6, we report the PPO hyperparameters used in the experiments. For each environment, all the hyperparameters are consistent across all the different curriculum strategies.

| Hyperparameters | POINTMASS-S | POINTMASS-D | BASICKAREL | BALLCATCHING | ANTGOAL |
|---|---|---|---|---|---|
| $N_{\text{steps}}$ | 1024 | 1024 | 2048 | 5120 | 1024 |
| $\gamma$ | 0.99 | 0.95 | 0.99 | 0.99 | 0.99 |
| $N_{\text{epochs}}$ | 10 | 10 | 10 | 10 | 10 |
| learning_rate | 3e−4 | 3e−4 | 3e−4 | 3e−4 | 2e−5 |
| batch_size | 64 | 64 | 64 | 64 | 32 |
| ent_coef | 0 | 0 | 0 | 0 | 5e−7 |
| clip_range | 0.2 | 0.2 | 0.2 | 0.2 | 0.1 |
| gae_lambda | 0.95 | 0.95 | 0.95 | 0.95 | 0.8 |
| max_grad_norm | 0.5 | 0.5 | 0.5 | 0.5 | 0.6 |
| vf_coef | 0.5 | 0.5 | 0.5 | 0.5 | 0.7 |

Figure 6: Different hyperparameters of the PPO method used in the experiments for each environment.

**Compute resources.** All the experiments were conducted on a cluster of machines with CPUs of model Intel Xeon Gold 6134M CPU @ 3.20GHz.

## C.3 Curriculum Strategies Evaluated

**Variants of the curriculum strategy.** Algorithm 2 provides a complete pseudo-code for the RL agent using PPO method when trained with PROCURL-VAL in the general setting of non-binary or dense rewards (see Section 3.3). In Eq. 1 and Algorithm 1, we defined $t$ at an episodic level; however, in Algorithm 2, $t$ denotes an environment step (in the context of the PPO method). For PROCURL-ENV, in line 24 of Algorithm 2, we estimate the probability of success for all the tasks using the additional rollouts obtained by executing the current policy in $\mathcal{M}$.

To achieve the constrained budget of evaluation steps in PROCURL-ENV$^{\text{x}}$ (with x $\in \{2, 4\}$), we reduce the frequency of updating $\text{PoS}_t$ since this is the most expensive operation for PROCURL-ENV requiring additional rollouts for each task. On the other hand, PROCURL-VAL updates $\text{PoS}_t$ by using the values

obtained from forward-pass on the critic model – this update happens whenever the critic model is updated (every 2048 training steps for BASICKAREL). This higher frequency of updating $\text{PoS}_t$ in PROCURL-VAL is why it is slower than PROCURL-ENV$^\text{x}$ (with $\text{x} \in \{2, 4\}$) for BASICKAREL. Note that the relative frequency of updates for POINTMASS is different in comparison to BASICKAREL because of very different pool sizes. Hence, the behavior in total clock times is different.

---

**Algorithm 2** RL agent using PPO method when trained with PROCURL-VAL in the general setting

---

1: **Input:** RL algorithm PPO, rollout buffer $\mathcal{D}$
2: **Hyperparameters:** policy update frequency $N_\text{steps}$, number of epochs $N_\text{epochs}$, number of minibatches $N_\text{batch}$, parameter $\beta$, $V_\text{min}$, and $V_\text{max}$
3: **Initialization:** randomly initialize policy $\pi_1$ and critic $V_1$; set normalized probability of success scores $\overline{V}_1(s) = 0$ and $\text{PoS}^*(s) = 1$, $\forall s \in \mathcal{S}_\text{init}$
4: **for** $t = 1, \ldots, T$ **do**
5:      // add an environment step to the buffer
6:      observe the state $s_t$, and select the action $a_t \sim \pi_t(s_t)$
7:      execute the action $a_t$ in the environment
8:      observe reward $r_t$, next state $s_{t+1}$, and done signal $d_{t+1}$ to indicate whether $s_{t+1}$ is terminal
9:      store $(s_t, a_t, r_t, s_{t+1}, d_{t+1})$ in the rollout buffer $\mathcal{D}$
10:      // choose new task when the current task/episode ends
11:      **if** $d_{t+1} = \texttt{true}$ **then**
12:          reset the environment state
13:          sample next task $s_{t+1}$ from $\mathbb{P}\big[s_{t+1} = s\big] \propto \exp\big(\beta \cdot \overline{V}_t(s) \cdot (1 - \overline{V}_t(s))\big)$
14:      // policy and $\overline{V}_t(s)$ update
15:      **if** $t \% N_\text{steps} = 0$ **then**
16:          set $\pi' \leftarrow \pi_t$ and $V' \leftarrow V_t$
17:          **for** $e = 1, \ldots, N_\text{epochs}$ **do**
18:              **for** $b = 1, \ldots, N_\text{batch}$ **do**
19:                  sample $b$-th minibatch of $N_\text{steps}/N_\text{batch}$ transitions $B = \{(s, a, r, s', d)\}$ from $\mathcal{D}$
20:                  update policy and critic using PPO algorithm $\pi', V' \leftarrow \text{PPO}(\pi', V', B)$
21:          set $\pi_{t+1} \leftarrow \pi'$ and $V_{t+1} \leftarrow V'$
22:          empty the rollout buffer $\mathcal{D}$
23:          // normalization for the environments with non-binary or dense rewards
24:          update $\overline{V}_{t+1}(s) \leftarrow \frac{V_{t+1}(s) - V_\text{min}}{V_\text{max} - V_\text{min}}$, $\forall s \in \mathcal{S}_\text{init}$ using forward passes on critic
25:      **else**
26:          maintain the previous values $\pi_{t+1} \leftarrow \pi_t$, $V_{t+1} \leftarrow V_t$, and $\overline{V}_{t+1} \leftarrow \overline{V}_t$
27: **Output:** policy $\pi_T$

---

**Hyperparameters of curriculum strategies.** In Figure 7, we report the hyperparameters of each curriculum strategy used in the experiments (for each environment). Below, we provide a short description of these hyperparameters:

1. $\beta$ parameter controls the stochasticity of the softmax selection.

2. $N_\text{pos}$ parameter controls the frequency at which $\overline{V}_t$ is updated. For PROCURL-ENV, we set $N_\text{pos}$ higher than $N_\text{steps}$ since obtaining rollouts to update $\overline{V}_t(s)$ is expensive. For all the other curriculum strategies, we set $N_\text{pos} = N_\text{steps}$. For SPACE, $N_\text{pos}$ controls how frequently the current task dataset is updated based on their curriculum. For SPDL, $N_\text{pos}$ controls how often we perform the optimization step to update the distribution for selecting tasks.

3. $c_\text{rollouts}$ determines the number of additional rollouts required to compute the probability of success score for each task (only for PROCURL-ENV).

4. $\{V_\text{min}, V_\text{max}\}$ are used in the environments with non-binary or dense rewards to obtain the normalized values $\overline{V}(s)$ (see Section 3.3). In Figure 7, $\{V_{\text{min},t}, V_{\text{max},t}\}$ denote the min-max values of the critic for states $\mathcal{S}_\text{init}$ at step $t$.

5. $\eta$ and $\kappa$ parameters as used in SPACE (Eimer et al., 2021).

6. $V_{\mathrm{LB}}$ performance threshold as used in SPDL (Klink et al., 2021).

7. $\rho$ staleness coefficient and $\beta_{\mathrm{PLR}}$ temperature parameter for score prioritization as used in PLR (Jiang et al., 2021b).

| Method | Hyperparameters | PointMass-s | PointMass-d | BasicKarel | BallCatching | AntGoal |
|---|---:|---:|---:|---:|---:|---:|
| ProCuRL-env | $\beta$ | 20 | 10 | 10 | 10 | 10 |
|  | $N_{\mathrm{pos}}$ | 5120 | 5120 | 102400 | 20480 | 81920 |
|  | $c_{\mathrm{rollouts}}$ | 20 | 20 | 20 | 20 | 20 |
|  | $\{V_{\min}, V_{\max}\}$ | n/a | $\{V_{\min,t}, V_{\max,t}\}$ | n/a | n/a | $\{0, 300\}$ |
| ProCuRL-val | $\beta$ | 20 | 10 | 10 | 10 | 10 |
|  | $N_{\mathrm{pos}}$ | 1024 | 1024 | 2048 | 5120 | 1024 |
|  | $\{V_{\min}, V_{\max}\}$ | n/a | $\{V_{\min,t}, V_{\max,t}\}$ | n/a | $\{0, 60\}$ | $\{0, 300\}$ |
| SPACE | $\eta$ | 0.1 | 0.1 | 0.5 | 0.1 | 0.1 |
|  | $\kappa$ | 1 | 1 | 64 | 1 | 1 |
|  | $N_{\mathrm{pos}}$ | 1024 | 1024 | 2048 | 5120 | 1024 |
| SPACE-alt | $\beta$ | 20 | 10 | 10 | 10 | 10 |
|  | $N_{\mathrm{pos}}$ | 1024 | 1024 | 2048 | 5120 | 1024 |
| SPDL | $V_{\mathrm{LB}}$ | 0.5 | 3.5 | 0.5 | 30 | 100 |
|  | $N_{\mathrm{pos}}$ | 1024 | 1024 | 2048 | 5120 | 1024 |
| PLR | $\rho$ | 0.5 | 0.9 | 0.9 | 0.7 | 0.3 |
|  | $\beta_{\mathrm{PLR}}$ | 0.1 | 0.3 | 0.1 | 0.3 | 0.1 |

Figure 7: We present the hyperparameters of the different curriculum strategies for all five environments. For SPDL, we choose the best performing $V_{\mathrm{LB}}$ in the non-binary environments from the following sets: set $\{1, \mathbf{3.5}, 10, 20, 30, 40\}$ for PointMass-D; set $\{20, 25, \mathbf{30}, 35, 42.5\}$ for BallCatching; set $\{50, \mathbf{100}, 200, 300, 400\}$ for AntGoal. For PLR, we choose the best performing pair $(\beta_{\mathrm{PLR}}, \rho)$ for each environment from the set $\{0.1, 0.3, 0.5, 0.7, 0.9\} \times \{0.1, 0.3, 0.5, 0.7, 0.9\}$.

## C.4 Additional Results

**Ablation and robustness experiments.** We conduct additional experiments to evaluate the robustness of ProCuRL-val w.r.t. different values of $\beta$ and different $\epsilon$-level noise in $V_t(s)$ values. The results are reported in Figure 8. Further, we conduct an ablation study on the form of our curriculum objective presented in Eq. 1. More specifically, we consider the following generalized variant of Eq. 1 with parameters $\gamma_1$ and $\gamma_2$:

$$s_t^{(0)} \quad \leftarrow \quad \underset{s \in \mathcal{S}_{\mathrm{init}}}{\arg\max} \left( \mathrm{PoS}_t(s) \cdot \left( \gamma_1 \cdot \mathrm{PoS}^*(s) - \gamma_2 \cdot \mathrm{PoS}_t(s) \right) \right) \tag{7}$$

In our experiments, we consider the following range of $\gamma_2/\gamma_1 \in \{0.6, 0.8, 1.0, 1.2, 1.4\}$. Our default curriculum strategy in Eq. 1 essentially corresponds to $\gamma_2/\gamma_1 = 1.0$. The results are reported in Figure 9.

**Performance on test set.** In Figure 10, we report the performance of the trained models in the training set and a test set for comparison purposes. For PointMass-S, we constructed a separate test set of 100 tasks by uniformly picking tasks from the task space. For BasicKarel, we have a train and test dataset of 24000 and 2400 tasks, respectively.

**Pool of harder tasks.** We sought to assess the effectiveness of ProCuRL-val on tasks where IID does not perform well. To demonstrate this, we construct a more challenging set of tasks for the PointMass-s environment. We generate half of these tasks by uniformly sampling over the context space. The remaining tasks are sampled from a bi-modal Gaussian distribution, where the means of the contexts [C-GatePosition, C-GateWidth] are $[-3, 1]$ and $[3, 1]$ for the two modes, respectively. In Figure 11, we present the results for ProCuRL-val and IID, and in Figure 12 the different distributions.

| Env  Method | POINTMASS-S | | | BASICKAREL | | |
|---|---|---|---|---|---|---|
| PROCURL-VAL | Performance | | | Performance | | |
| | 0.25M | 0.5M | 1M | 0.25M | 0.5M | 1M |
| $\beta = 10$ | $0.48 \pm 0.17$ | $0.58 \pm 0.19$ | $0.70 \pm 0.18$ | $0.06 \pm 0.03$ | $0.30 \pm 0.08$ | $0.71 \pm 0.05$ |
| $\beta = 15$ | $0.42 \pm 0.17$ | $0.64 \pm 0.17$ | $0.74 \pm 0.15$ | $0.12 \pm 0.04$ | $0.38 \pm 0.04$ | $0.71 \pm 0.05$ |
| $\beta = 20$ | $0.48 \pm 0.15$ | $0.64 \pm 0.17$ | $0.71 \pm 0.18$ | $0.18 \pm 0.06$ | $0.42 \pm 0.06$ | $0.75 \pm 0.06$ |
| $\beta = 25$ | $0.45 \pm 0.18$ | $0.60 \pm 0.19$ | $0.65 \pm 0.21$ | $0.22 \pm 0.03$ | $0.38 \pm 0.04$ | $0.62 \pm 0.05$ |
| $\beta = 30$ | $0.54 \pm 0.18$ | $0.64 \pm 0.20$ | $0.74 \pm 0.19$ | $0.20 \pm 0.06$ | $0.36 \pm 0.07$ | $0.67 \pm 0.07$ |
| $\epsilon = 0.00$ | $0.48 \pm 0.15$ | $0.64 \pm 0.17$ | $0.71 \pm 0.18$ | $0.06 \pm 0.03$ | $0.30 \pm 0.08$ | $0.71 \pm 0.05$ |
| $\epsilon = 0.01$ | $0.53 \pm 0.18$ | $0.62 \pm 0.19$ | $0.71 \pm 0.20$ | $0.06 \pm 0.02$ | $0.30 \pm 0.06$ | $0.69 \pm 0.04$ |
| $\epsilon = 0.05$ | $0.39 \pm 0.16$ | $0.60 \pm 0.17$ | $0.70 \pm 0.20$ | $0.06 \pm 0.02$ | $0.31 \pm 0.06$ | $0.72 \pm 0.04$ |
| $\epsilon = 0.1$ | $0.47 \pm 0.17$ | $0.59 \pm 0.16$ | $0.67 \pm 0.18$ | $0.06 \pm 0.03$ | $0.30 \pm 0.07$ | $0.69 \pm 0.07$ |
| $\epsilon = 0.2$ | $0.49 \pm 0.16$ | $0.61 \pm 0.18$ | $0.68 \pm 0.18$ | $0.04 \pm 0.02$ | $0.26 \pm 0.08$ | $0.74 \pm 0.03$ |

Figure 8: Robustness of PROCURL-VAL w.r.t. different values of $\beta$ and different $\epsilon$-level noise in $V_t(s)$ values. We present the results for the POINTMASS-S environment and BASICKAREL environment. We report the mean reward ($\pm$ $t \times$ standard error, where $t$ is the value from the t-distribution table for 95% confidence) at 0.25, 0.5, and 1 million training steps averaged over 20 and 10 random seeds, respectively.

| Env  Method | POINTMASS-S | | | BASICKAREL | | |
|---|---|---|---|---|---|---|
| PROCURL-VAL | Performance | | | Performance | | |
| | 0.25M | 0.5M | 1M | 0.25M | 0.5M | 1M |
| $\gamma_2/\gamma_1 = 0.6$ | $0.33 \pm 0.14$ | $0.50 \pm 0.14$ | $0.55 \pm 0.13$ | $0.08 \pm 0.04$ | $0.21 \pm 0.07$ | $0.41 \pm 0.08$ |
| $\gamma_2/\gamma_1 = 0.8$ | $0.26 \pm 0.15$ | $0.43 \pm 0.17$ | $0.55 \pm 0.20$ | $0.11 \pm 0.03$ | $0.36 \pm 0.05$ | $0.64 \pm 0.07$ |
| $\gamma_2/\gamma_1 = 1.0$ | $0.48 \pm 0.15$ | $0.64 \pm 0.17$ | $0.71 \pm 0.18$ | $0.06 \pm 0.03$ | $0.30 \pm 0.08$ | $0.71 \pm 0.05$ |
| $\gamma_2/\gamma_1 = 1.2$ | $0.42 \pm 0.19$ | $0.55 \pm 0.19$ | $0.65 \pm 0.17$ | $0.07 \pm 0.04$ | $0.30 \pm 0.11$ | $0.72 \pm 0.08$ |
| $\gamma_2/\gamma_1 = 1.4$ | $0.39 \pm 0.16$ | $0.59 \pm 0.17$ | $0.59 \pm 0.15$ | $0.04 \pm 0.02$ | $0.21 \pm 0.08$ | $0.71 \pm 0.06$ |

Figure 9: Performance comparison of the generalized form of our curriculum strategy presented in Eq. 7 w.r.t. different values of $\gamma_2/\gamma_1$. We present the results for the POINTMASS-S environment and BASICKAREL environment. We report the mean reward ($\pm$ $t \times$ standard error, where $t$ is the value from the t-distribution table for 95% confidence) at 0.25, 0.5, and 1 million training steps averaged over 20 and 10 random seeds, respectively.

| Env
Method | POINTMASS-S | | BASICKAREL | |
|---|---|---|---|---|
| | Performance (1M) | | Performance (2M) | |
| | Train Set | Test Set | Train Set | Test Set |
| PROCURL-ENV | 0.84 | 0.78 | 0.92 | 0.90 |
| PROCURL-VAL | 0.71 | 0.65 | 0.91 | 0.90 |
| SPACE | 0.34 | 0.28 | 0.65 | 0.64 |
| SPACE-ALT | 0.47 | 0.40 | 0.82 | 0.81 |
| SPDL | 0.55 | 0.48 | 0.88 | 0.87 |
| PLR | 0.69 | 0.60 | 0.88 | 0.88 |
| IID | 0.39 | 0.32 | 0.90 | 0.89 |

Figure 10: Performance of the curriculum strategies, discussed in Section 4.2, in the training set and a test set. We report the performance, i.e., expected mean reward, of the best model obtained during training for all the methods. The training steps to achieve this performance is shown in parenthesis for each environment (M is $10^6$ steps). We present the results for the POINTMASS-S environment and BASICKAREL environment and report the mean reward averaged over 20 and 10 random seeds, respectively.

| Env
Method | POINTMASS-S | | | | |
|---|---|---|---|---|---|
| | Performance | | | | |
| | 0.25M | 0.5M | 1M | 1.5M | 2M |
| PROCURL-VAL | $0.07 \pm 0.07$ | $0.19 \pm 0.11$ | $0.40 \pm 0.15$ | $0.46 \pm 0.16$ | $0.49 \pm 0.16$ |
| IID | $0.01 \pm 0.01$ | $0.03 \pm 0.03$ | $0.05 \pm 0.06$ | $0.04 \pm 0.04$ | $0.03 \pm 0.03$ |

Figure 11: Performance comparison of our curriculum strategy, PROCURL-VAL, and IID in a pool of harder tasks for the POINTMASS-S environment. We report the mean reward ($\pm$ $t\times$standard error, where $t$ is the value from the t-distribution table for 95% confidence) at 0.25, 0.5, 1, 1.5 and 2 million training steps averaged over 20 random seeds.

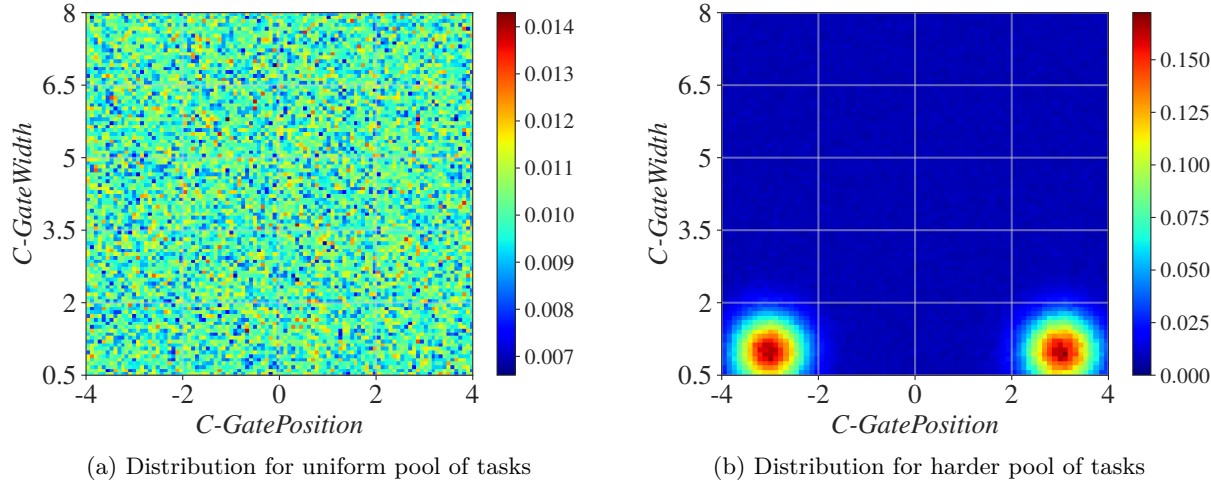

(a) Distribution for uniform pool of tasks        (b) Distribution for harder pool of tasks

Figure 12: **(a)** shows the distribution of context values used to generate the uniform pool of tasks for the main experiments **(b)** the distribution of context values that is used to generate a harder pool of tasks.

