# OpenReview forum: "Proximal Curriculum for Reinforcement Learning Agents"
_TMLR — Accepted by TMLR_

### Review · Reviewer_jtz6 · 2022-12-28

**Summary Of Contributions:**

This paper proposes a method to use curriculum learning to accelerate reinforcement learning (RL). Intuitively, the proposed method samples tasks that are more difficult than those that the RL agent has currently mastered, but not too much more difficult. More precisely, tasks are defined by different initial states, each of which is evaluated using a learned value function. The proposed method samples initial states that both (1) have a large value, and (2) have a large regret relative to the optimal policy. The proposed method is straightforward to implement and achieves good results on benchmark goal-reaching tasks from prior work. The paper also provides some analysis arguing why the proposed method is theoretically justified.

**Audience:**

Yes

**Broader Impact Concerns:**

This paper doesn't have direct broader impact concerns. That said, it might be good to at least mention the potential for RL to have large positive/negative societal impacts

**Claims And Evidence:**

Yes

**Requested Changes:**

High-level comments
* Can we interpret the selection criterion PoS(s) x (PoS*(s) - PoS(s)) as the geometric mean of (1) the probability of solving the task and (2) the expected regret on this task? These are both natural quantities to maximize (w.r.t. curriculum learning). If so, it might be good to include this intuition.
* "at the goal states" -- How does this work in continuous environments, where "at" isn't well defined?
* Eq. 6 -- Does this amount to a Gaussian distribution over _values_? In expectation, do the sampled tasks have a probability-of-success of $\frac{1}{2}(PoS(s) + 1)$?
* Given that all the tasks are sparse goal-reaching problems, I'd recommend using an RL algorithm designed for that setting (e.g., [1,2,3]). This will _decrease_ the assumptions needed by the method, as it will no longer require manually-defining a function to say whether you've gotten close enough to the goal. [See comment about "at the goal states"]

Minor writing comments
* "Recent works have" -- Cite.
* Check citep vs citet. E.g., "in both (Florensa" should use citet.
* "The authors in (...) propose" -> "... propose"
* Notation -- I'd recommend using $s^{(i)}$ instead of $s^{(\tau)}$, as some readers will expect that $\tau$ is the trajectory ($\xi$) rather than an index.
* Footnote 1 -- I'd recommend writing this out in words, using an example. For readers not familiar with this construction, the notation can seem obtuse.
* sample efficient manner" -- Check grammar here. I think "manner" should be plural, or an article should be added.
* Fig 1b -- I'd recommend showing a real rendering of the BallCatching environment.
* Fig 3 -- Tables with many numbers are hard to read. I'd recommend replacing this with a bar plot, and putting the table in the appendix.
* Fig 4, 5 -- I found it hard to interpret these figures. I'd recommend expanding the captions to provide some intuition. E.g., what exactly do the Y axes correspond to, what's the intuition for why they should be large at some time and small at other times?

-----------------

[1] https://proceedings.neurips.cc/paper/2019/file/c8d3a760ebab631565f8509d84b3b3f1-Paper.pdf

[2] https://arxiv.org/pdf/2206.07568.pdf

[3] https://arxiv.org/pdf/1905.07866.pdf


**Strengths And Weaknesses:**

Strengths
* Simplicity -- It seems straightforward to implement the method. I like the idea of using the learned value function to estimate the probability of success.
* The writing is generally clear, and the explanation of the prior methods is good.


Weaknesses
* The tasks in the experiments are pretty easy; it'd be great to include at least one really compelling task that cannot be solved without the curriculum.
* I wasn't able to understand the theoretical justification. I was hoping to see a result that said something like "selecting tasks using the proposed method decreases the cumulative regret from O(stuff) to O(stuff), as compared with iid task selection." As is, Theorem 1 doesn't mention any objective criterion; Sec. 3.2 mentions differences in parameters (Eq. 2), but that is never precisely related to differences in values (the Lipschitz constant L isn't assumed) nor to the update rule in Sec. 3.2.1.

---

> ### Author Response · Authors · 2023-03-11
> **Response to Reviewer jtz6**
>
> Thank you for carefully reviewing our paper! We greatly appreciate your feedback. Please see below our responses to your comments.
>
> -----
>
> **1. The tasks in the experiments are pretty easy; it'd be great to include at least one really compelling task that cannot be solved without the curriculum.**
>
> We have conducted our experiments on the *same* set of environments used in the state-of-the-art baselines (SPDL and SPaCE) for the pool-based setting with the uniform performance objective. Our goal (similar to SPDL and SPaCE) is to train a policy that can perform uniformly well across all tasks in the task space, which is a different objective from other curriculum-based approaches where the focus is on performance with respect to a *target distribution* (such as a concentrated distribution on complex tasks). In such scenarios, including unsolvable tasks in the task space can be meaningful. While we recognize the potential benefits of extending our approach to target task settings, we have left this as a natural direction for future work.
>
> -----
>
> **2. Motivation for the theoretical justification section.**
>
> The objective of our theoretical justification section is not to provide convergence analysis; although it is certainly an interesting theoretical avenue to explore. Instead, our aim is to demonstrate that our curriculum optimization objective can be seen as a greedy one-step optimization of the learning agent's training objective in two simple learning settings.
>
> We have revised the paper to simplify the presentation of the theoretical section. The updated theorems now clearly illustrate how our curriculum strategy relates to the expected improvement in the training objective. We have also moved the analysis of the abstract agent to the Appendix to enhance the readability of Section 3.2.
>
> -----
>
> **3. Can we interpret the selection criterion $\text{PoS}(s) \times (\text{PoS}^\star(s) - \text{PoS}(s))$ as the geometric mean of (1) the probability of solving the task and (2) the expected regret on this task? These are both natural quantities to maximize (w.r.t. curriculum learning). If so, it might be good to include this intuition.**
>
> We would like to thank the reviewer for their insightful intuition. In the revised version of the paper, we have included this interpretation in Section 3.1.
>
> -----
>
> **4. "at the goal states" -- How does this work in continuous environments, where "at" isn't well defined?**
>
> Defining success by the proximity to a goal location is a prevalent approach in the literature for continuous goal-based environments. We have adopted this same strategy for the goal-based environments of the experimental section.
>
> -----
>
> **5. Eq. 6 -- Does this amount to a Gaussian distribution over values?**
>
> Eq. 6 is simply a softmax version of Eq. 1, and it does not necessarily amount to a Gaussian distribution over values.
>
> -----
>
> **6. Given that all the tasks are sparse goal-reaching problems, I'd recommend using an RL algorithm designed for that setting (e.g., [1,2,3]).**
>
> In our work, we want to keep the environments and learning agents consistent with the baseline methods (SPDL, and SPaCE) for fair comparison; it is indeed an interesting direction to empirically investigate the effect of a curriculum on these goal-based RL agents.
>
> -----
>
> **7. Minor Comments**
>
> We thank the reviewer for pointing out these minor comments. We have incorporated most of them in the revised version of the paper.
>
> -----

---

> > ### Comment · Reviewer_jtz6 · 2023-03-11
> > **Reviewer response**
> >
> > Dear authors,
> >
> > Thanks for the detailed response. The revisions address my concern about the theoretical result.
> >
> > For the experiments, thanks for clarifying the exact problem setting and for clarifying that these are the same tasks as used in prior work (this is a great reason to use them!). I wonder whether it'd nonetheless be possible to include some really compelling example for curriculum learning. For example, some set of tasks where iid sampling would only result in solving 10% of the tasks, but the proposed curriculum learning method solves 90% of the tasks.
> >
> > > "at the goal states" -- How does this work in continuous environments, where "at" isn't well defined?
> >
> > To clarify, this means that the proposed method requires a "proximity function" or "distance function"? It might be good to mention this assumption, as well as citations to prior work that makes similar assumptions.

---

> > > ### Author Response · Authors · 2023-03-15
> > > **Follow-up response to Reviewer jtz6**
> > >
> > > We thank the reviewer for acknowledging our response and the changes made in the revised paper.
> > >
> > > -----
> > >
> > > **A compelling example of a pool of tasks where the proposed curriculum learning method outperforms IID significantly.**
> > >
> > > Conducting an experiment with a complex pool of tasks where our method succeeds while the IID approach fails is certainly intriguing. To demonstrate that our strategy can succeed even when IID fails, we did an initial experiment in this direction as follows. We constructed a pool of harder tasks for the PointMass-s environment, and ran experiments using IID and ProCuRL-val. We have now uploaded a new revised version of the paper. We refer the reviewer to Figure 11 and the discussion in Appendix C.4, where we have included further details about the pool construction and the results of this experiment.
> > >
> > > -----
> > >
> > > **The proposed method requires a "proximity function" or "distance function"? It might be good to mention this assumption, as well as citations to prior work that makes similar assumptions.**
> > >
> > > The proposed curriculum learning method is applied to goal-based environments, where it is common for these environments to have a distance function to measure goal proximity. However, as suggested by the reviewer, it would be interesting to explore the impact of our curriculum strategy on goal-based reinforcement learning algorithms that do not assume a distance or proximity function needs to be specified. In the new revised version of the paper, we have added a remark at the end of Section 4.1 addressing this point and included the references mentioned by the reviewer.
> > >
> > > -----

---

> > > > ### Comment · Reviewer_jtz6 · 2023-03-16
> > > > **Reviewer response**
> > > >
> > > > Dear authors,
> > > >
> > > > These revisions address my remaining concerns about the paper! I will advocate for accepting the paper.
> > > >
> > > > Nitpick: It might be good to add a small visualization of goal distribution used for the new Fig. 11.

---

### Review · Reviewer_Brna · 2023-03-01

**Summary Of Contributions:**

The paper introduces an approach for the automated generation of curricula based on the idea of a Zone of Proximal Development, intuitively aiming to select tasks that are neither too hard nor too easy. The authors formulate this selection as the product between current agent performance and regret. This particular choice is motivated by theoretical investigations in two examples, in which the authors can show that their strategy leads to an optimal improvement of overall success.

The authors then carry on to compare their proposed methods against different baselines, showing good performance.

All in all, I agree with the contributions that the authors claim to make in this paper.

**Audience:**

Yes

**Claims And Evidence:**

Yes

**Requested Changes:**

I would like to ask the authors to clarify open questions (performance difference of SPDL for PointMass-S, finite-time horizon $H$, importance of Section $3.2.1$, number of seeds used for evaluation). Apart from this, I would like to see a discussion of ways to extend the current theoretical analysis.
Finally, I think that making the results of Theorem 2 more intuitive (e.g. via visualizations) as well as discussing the influence of $p_{\text{rand}}(s)$ would give additional insights.

**Strengths And Weaknesses:**

The paper is well-written and easy to follow. The authors discuss the influence of hyperparameter choices and provide proof for their theoretical results in the appendix. The authors even evaluate an additional

Regarding weaknesses, I have a few minor points:
 * I already reviewed this paper for ICLR 2022 (where it did not get accepted). Back then, a reviewer proposed a different implementation of SPDL that improved performance. While the results mostly agree with the results reported by the reviewer back then, there seems to be a rather strong performance difference for SPDL in the PointMass-S environment. The reviewer reported an average performance of $0.74 \pm 0.03$ computed from 10 seeds whereas the results here show only a performance of $0.52 \pm 0.08$. Can the reviewers explain where this strong difference comes from?
 * The extra time parameter $H$ is a bit confusing to me. For finite-horizon MDPs, it is typically important to provide the time step as part of the state. I cannot find this time index being passed to e.g. the policy in the notation. Is this because the time step is assumed to be contained in the state implicitly? If so, I'd like to ask the authors to clearly state this detail.
 * An inherent weakness of the theoretical investigations is that the authors do not allow parameter (i.e. information) sharing between contexts. While I understand why this is done, I would like the authors to discuss this limitation a bit more in the paper. What could be the next steps for incorporating the aspect of information sharing in their theory? Such a discussion would greatly help people in pushing the theoretical investigations further.
 * Before Equation (2), the authors could provide a reference to the paper by Graves et al. (already in the references) and make clear that Equation (2) is an approximation to what Graves et al. refer to as expected learning progress.
 * I am wondering whether Section 3.2.1 is beneficial to the overall paper. What I dislike about this section is that a hypothesis is made about the learning rule and the agent parameters are detached from the actual behavior. Section 3.2.2 seems to offer the same insights while having a much clearer motivation (as a contextual bandit problem) and using the REINFORCE algorithm. Could the authors discuss why they would like to keep Section 3.2.1? Or would they be willing to move this section to the appendix and instead provide more details for the results in Section 3.2.2?
 * An improvement for Section 3.2.2. could e.g. be to visualize the quantities that are shown in Theorem 2 and further discuss how $\max_p C_t(p_{\text{rand}}(s), p)$ begins to differ from $\max_p p \cdot (1 - p)$ when $p_{\text{rand}}(s)$ increasingly differs from 1.
 * The caption of Figure 2 left me a bit puzzled as to how many seeds were used for which environment. Could the authors edit this caption to clearly say how many seeds were used for which environment?

---

> ### Author Response · Authors · 2023-03-11
> **Response to Reviewer Brna**
>
> Thank you for carefully reviewing our paper! We greatly appreciate your feedback. Please see below our responses to your comments.
>
> -----
>
> **1. Performance difference for SPDL in the PointMass-S environment.**
>
> The SPDL results presented in our paper are based on the proposed implementation of the previous reviewer, and we reran all of the experiments. However, for the PointMass-S environment, we observed that different training seeds (runs) could result in significantly different performance outcomes due to its large variance. Therefore, we conducted experiments for 20 seeds and are reporting these results.
>
> -----
>
> **2. Confusion regarding the finite-horizon MDPs with maximum episode length $H$.**
>
> As pointed out by the reviewer, when dealing with general finite-horizon MDPs, it is important to include the time step as part of the state. However, to avoid complicating the notation with additional indexing, we have assumed that the time step is implicitly included in the state. We have incorporated this detail in the revised version of the paper. In Section 3.2, we analyze contextual bandit tasks for which stationary policies are sufficient. Further, note that our final practical curriculum strategies can also be applied to discounted MDPs.
>
> -----
>
> **3. What could be the next steps for incorporating the aspect of information sharing in the theoretical analysis?**
>
> Incorporating correlated task settings into our analysis of the curriculum strategy would add significant value to the theoretical framework in addition to our current focus on independent task settings. A potential approach to achieve this could be to use a distance metric over the context space, which would facilitate the development of a more generalized version of our curriculum strategy. We have included this point in Section 5 of the revised version of our paper.
>
> -----
>
> **4. Are the authors willing to move the analysis of the abstract agent to the appendix for better readability?**
>
> Based on the reviewer's suggestion, we have moved the analysis of the abstract agent to the Appendix to enhance the readability of Section 3.2. We would like to emphasize that the motivation behind the abstract agent is to design an update rule that closely resembles a policy gradient style update while encapsulating the idea of learning progress with tasks that lie in the zone of proximal development (ZPD).
>
> -----
>
> **5. Illustration of the quantities in (previous) Theorem 2 to make it more clear.**
>
> In the revised version of the paper, we have updated the theorem statements for the REINFORCE agent and the abstract agent. The current versions of these theorems now clearly illustrate the relationship between our curriculum strategy and the expected improvement in the training objective. Additionally, it is evident from the updated theorem statements how the value of $\max_s C_t(s)$ starts to deviate from $\max_s p(s) \cdot (1-p(s))$ as $p_\textnormal{rand}(s)$ moves away from 1.
>
> -----
>
> **6. Improve the clarity of the caption in Figure 2.**
>
> Thank you for your input. We have revised the caption of Figure 2 to indicate the number of seeds used for each environment specifically.
>
> -----

---

### Review · Reviewer_F5vg · 2023-03-02

**Summary Of Contributions:**

The paper proposes a new method for curriculum learning that selects tasks (in this work initial states) such that are neither too easy nor too hard. The proposed strategy roughly amounts to choosing tasks that maximise ```current success rate * (best success rate - current success rate)```.  The paper is inspired by the concept of "zone of proximal development" from the educational psychology literature.

The paper provides some theoretical results to motivate the proposed method as well as an empirical evaluation on a number of different environments.

**Audience:**

Yes

**Broader Impact Concerns:**

No concerns.

**Claims And Evidence:**

No

**Requested Changes:**

* Please clarify why $C_t(p^*, p)$ is differentiable in your examples (or in general)
* Please justify the choice of baselines.

**Strengths And Weaknesses:**

Strengths:
* Using the idea of a ZPD is well motivated.
* Empirical evaluation compares a number of variants of the proposed method (Monte Carlo estimates of success vs a function approximators) and relevant baselines across a decent number of environments. The proposed methods performs well.

Weaknesses:
* I have concerns about the theoretical results that serve as motivation. Specifically, it seems to me that there is no guarantee in general (and certainly not in discrete environments) that the set $\mathcal{D}_t(p^*, p)$ is non empty for most p. Certainly the differentiability of $C_t(p^*, p)$ depends on $\mathcal{D}_t(p^*, q)$ being non-empty values of q neighbourhood of $p$.
* I find the theoretical results hard to read and the notation for e.g. $\theta$ confusing. I think the writing in this section could be improved. I would also not call $\theta$ a knowledge parameter.
* I'm also not convinced that the theoretical results sufficiently elucidate the method to be worth including even if correct. Eg. the method drops $PoS*$ entirely. This also raises the question of what happens in situations where some tasks are very hard or impossible.
* In the empirical section I wonder about the set of baselines. What is the reason for not comparing to prioritized level replay [1] adapted to this setting? To my knowledge PLR is a state-of-the-art curriculum learning method that has been shown to scale to quite large domains [2]

[1] Jiang, Minqi, Edward Grefenstette, and Tim Rocktäschel. "Prioritized level replay." International Conference on Machine Learning. PMLR, 2021.

[2] Team, Adaptive Agent, et al. "Human-Timescale Adaptation in an Open-Ended Task Space." arXiv preprint arXiv:2301.07608 (2023).

---

> ### Author Response · Authors · 2023-03-11
> **Response to Reviewer F5vg**
>
> Thank you for carefully reviewing our paper! We greatly appreciate your feedback. Please see below our responses to your comments.
>
> -----
>
> **1. Non-emptiness of $D_t(p^\star,p)$ and differentiability of $C_t(p^\star,p)$.**
>
> We thank the reviewer for pointing out this issue. As the reviewer noted, the set $D_t(p^\star,p)$ is not guaranteed to be non-empty for most $p \in [0,1]$, for the discrete pool of tasks considered in our analysis. Consequently, $C_t(p^\star, p)$ cannot be differentiated. Nevertheless, most proof derivations for Theorems 1 and 2 remain unchanged, except for the partial derivative derivations at the end of the proofs, which have been removed. We have revised Section 3.2 and the theorem statements to reflect these updates in the revised version of the paper.
>
> -----
>
> **2. Readability of the theoretical section**
>
> The modifications made to address Q.1 have already led to a simplified presentation of the theoretical section. Additionally, we have moved the analysis of the abstract agent to the Appendix to improve the readability of Section 3.2.
>
> -----
>
> **3. Prioritized Level Replay [1] adapted to this setting as a baseline?**
>
> We thank the reviewer for bringing this up. While PLR was originally designed for the procedurally generated content setting, we have successfully adapted its implementation to our pool-based setting and included it as an additional baseline in our study. The revised version of the paper includes the updated results (please refer to Figures 2 and 3).
>
> -----
>
> **4. the method drops $PoS^\star$ entirely … what happens in situations where some tasks are very hard or impossible.**
>
> The reviewer has made an interesting point regarding the presence of hard or impossible tasks in the task space. Even if we set $\textnormal{PoS}^\star=1$ for all tasks, the curriculum strategy remains highly effective. This is because it avoids selecting unsolvable tasks, as their value with respect to Eq. 1 is equal to 0 when $\textnormal{PoS}_t=0$.
>
> -----

---

### Decision · Action_Editors · 2023-03-31

**Recommendation:** Accept as is

**Comment:**

The reviewers had an open discussion with the authors, and any issues were addressed there. The recommendations from the reviewers had no additional information, simply stating that the authors had answered their questions and supported the work being accepted.

Let me suggest one comment to minorly improve the work.

It would be useful to more clearly highlight at least in the table, in Figure 3, if the differences are significant. Standard errors can be a bit misleading, and it is likely better to at least include 95% confidence intervals for a student-t distribution. Here that effectively means multiplying the standard error by 2. I understand that this does not change much (and so does not seem to be pointful), but it does at least reflect the number of runs used for each environment. For example, an experiment with only 3 runs would have a notably larger multiplier.

I am accepting this paper without this as a required revision, since it can mostly be inferred from the current table. But nonetheless I encourage the authors to consider 95% student-t intervals as a bit more informative than simply standard errors, and to justify why it is reasonably to report student-t intervals (already, those are also making strong assumptions too).

**Audience:**

Yes.

**Claims And Evidence:**

The reviewers agreed that the papers supports it claims with clear evidence.

One extra point: The paper seems to fairly treat the hyperparameters for the new approach, and allows a more systematic sweep for competitors, which is nice to see.